# Robust, fast and accurate mapping of diffusional mean kurtosis

**Megan E Farquhar[1], Qianqian Yang[1,2,3]\*, Viktor Vegh[4,5]**

[1]School of Mathematical Sciences, Faculty of Science, Queensland University of Technology, Brisbane, Australia; [2]Centre for Data Science, Queensland University of Technology, Brisbane, Australia; [3]Centre for Biomedical Technologies, Queensland University of Technology, Brisbane, Australia; [4]Centre for Advanced Imaging, The University of Queensland, Brisbane, Australia; [5]ARC Training Centre for Innovation in Biomedical Imaging Technology, Brisbane, Australia

**\*For correspondence:**
q.yang@qut.edu.au

**Competing interest:** The authors declare that no competing interests exist.

**Abstract** Diffusional kurtosis imaging (DKI) is a methodology for measuring the extent of non-Gaussian diffusion in biological tissue, which has shown great promise in clinical diagnosis, treatment planning, and monitoring of many neurological diseases and disorders. However, robust, fast, and accurate estimation of kurtosis from clinically feasible data acquisitions remains a challenge. In this study, we first outline a new accurate approach of estimating mean kurtosis via the sub-diffusion mathematical framework. Crucially, this extension of the conventional DKI overcomes the limitation on the maximum *b*-value of the latter. Kurtosis and diffusivity can now be simply computed as functions of the sub-diffusion model parameters. Second, we propose a new fast and robust fitting procedure to estimate the sub-diffusion model parameters using two diffusion times without increasing acquisition time as for the conventional DKI. Third, our sub-diffusion-based kurtosis mapping method is evaluated using both simulations and the Connectome 1.0 human brain data. Exquisite tissue contrast is achieved even when the diffusion encoded data is collected in only minutes. In summary, our findings suggest robust, fast, and accurate estimation of mean kurtosis can be realised within a clinically feasible diffusion-weighted magnetic resonance imaging data acquisition time.

## eLife assessment

This paper proposes a **valuable** new method for the assessment of the mean kurtosis for diffusional kurtosis imaging by utilizing a recently introduced sub-diffusion model. The evidence supporting the claims that this technique is robust and accurate in brain imaging is **solid**; however, there is a need to include a summary of the clear limitations.

## Introduction

Diffusion-weighted magnetic resonance imaging (DW-MRI) over a period of more than 30 years has become synonymous with tissue microstructure imaging. Measures of how water diffuses in heterogeneous tissues allow indirect interpretation of changes in tissue microstructure (*Le Bihan and Johansen-Berg, 2012*). DW-MRI has predominantly been applied in the brain, where properties of white matter connections between brain regions are often studied (*Lebel et al., 2019*), in addition to mapping tissue microstructural properties (*Tournier, 2019*). Applications outside of the brain have clinical importance as well, and interest is growing rapidly.

Generally, DW-MRI involves the setting of diffusion weightings and direction over which diffusion is measured. Whilst diffusion tensor imaging (DTI) can be performed using a single diffusion weighting,

a so-called b-shell, and at least six diffusion encoding directions (*Le Bihan et al., 2001*), other models tend to require multiple b-shells each having multiple diffusion encoding directions. Diffusional kurtosis imaging (DKI) is a primary example of a multiple b-shell, multiple diffusion encoding direction method (*Jensen et al., 2005*). DKI is considered as an extension of DTI (*Jensen and Helpern, 2010*; *Hansen et al., 2013*; *Veraart et al., 2011b*), where the diffusion process is assumed to deviate away from standard Brownian motion, and the extent of such deviation is measured via the kurtosis metric. Essentially, the increased sampling achieved via DKI data acquisitions allows more complex models to be applied to data (*Van Essen et al., 2013*; *Shafto et al., 2014*), in turn resulting in metrics of increased utility for clinical decision making.

Recent clinical benefits of using kurtosis metrics over other DW-MRI-derived measures have been demonstrated for grading hepatocellular carcinoma (*Li et al., 2022b*), prognosing chronic kidney disease (*Liu et al., 2022*), differentiating parotid gland tumours (*Huang et al., 2022*), measuring response to radiotherapy treatment in bone tumour (*Guo et al., 2022b*) and glioblastoma (*Goryawala et al., 2023*), identifying tissue abnormalities in temporal lobe epilepsy patients with sleep disorders (*Guo et al., 2022a*) and brain microstructural changes in mild traumatic brain injury (*Wang et al., 2022*), monitoring of renal function and interstitial fibrosis (*Li et al., 2022a*), detecting the invasiveness of bladder cancer into muscle (*Li et al., 2022d*), aiding management of patients with depression (*Maralakunte et al., 2023*), delineating acute infarcts with prognostic value (*Hu et al., 2022*), predicting breast cancer metastasis (*Zhou et al., 2023*), diagnosing Parkinson's disease (*Li et al., 2022c*), amongst others reported prior and not listed here.

The routine use of DKI in the clinic has nonetheless lagged due the inability to robustly estimate the kurtosis metric (*Veraart et al., 2011a*; *Tabesh et al., 2011*; *Kuder et al., 2012*; *Henriques et al., 2021*). A known requirement for estimating kurtosis in DKI is to restrict the maximum b-value to $2000 - 3000$ s/mm$^2$ for brain studies (*Jensen et al., 2005*; *Jensen and Helpern, 2010*; *Poot et al., 2010*), with the optimal maximum b-value found to be dependent on tissue type (*Poot et al., 2010*). This suggests that the traditional kurtosis model is less accurate at representing the diffusion signal at large b-values. Moreover, multiple b-shell, multiple direction high-quality DW-MRI data can take many minutes to acquire, which poses challenges for clinical imaging protocols involving a multitude of MRI contrasts already taking tens of minutes to execute. Reduction of DKI data acquisition times through parallel imaging, optimisation of b-shells and directions have been investigated (*Zong et al., 2021*; *Heidemann et al., 2010*; *Zelinski et al., 2008*), and DW-MRI data necessary for DKI analysis has been shown to supersede the data required for DTI (*Veraart et al., 2011b*). Therefore, an optimised DKI protocol can potentially replace clinical DTI data acquisitions without adversely affecting the estimation of DTI metrics.

For DKI to become a routine clinical tool, DW-MRI data acquisition needs to be fast and provides a robust estimation of kurtosis. The ideal protocol should have a minimum number of b-shells and diffusion encoding directions in each b-shell. The powder averaging over diffusion directions improves the signal-to-noise ratio (SNR) of the DW-MRI data used for parameter estimation. Whilst this approach loses out on directionality of the kurtosis, it nonetheless provides a robust method of estimating mean kurtosis (*Henriques et al., 2021*), a metric of significant clinical value.

Instead of attempting to improve an existing model-based approach for kurtosis estimation, as has been considered by many others, we considered the problem from a different perspective. In view of the recent generalisation of the various models applicable to DW-MRI data (*Yang et al., 2022*), the sub-diffusion framework provides new, unexplored opportunities, for fast and robust kurtosis mapping. We report on our investigation into the utility of the sub-diffusion model for practically useful mapping of mean kurtosis.

## Results

Simulation studies were conducted to establish the requirements on the number of diffusion times and the separation between them for accurate estimation of mean kurtosis based on the sub-diffusion model augmented with random Gaussian noise, following (*Equation 10*). Testing and validation were performed using the human Connectome 1.0 brain dataset (*Tian et al., 2022*). The $2 \times 2 \times 2$ mm$^3$ resolution data were obtained using two diffusion times ($\Delta = 19, 49$ ms) with a pulse duration of $\delta = 8$ ms and $G = 31, 68, 105, 142, 179, 216, 253$, and 290 mT/m, respectively, generating b-values = 50, 350, 800, 1500, 2400, 3450, 4750, and 6000 s/mm$^2$, and b-values = 200, 950, 2300, 4250, 6750,

9850, 13500, and 17800 s/mm², according to $b$-value $= (\gamma\delta G)^2(\Delta - \delta/3)$. Up to 64 diffusion encoding directions per b-shell were set. The traditional method for mean kurtosis estimation was implemented (producing $K_{DKI}$), which is limited to the use of DW-MRI generated using a single diffusion time (*Jensen et al., 2005*; *Jensen and Helpern, 2010*; *Veraart et al., 2011a*; *Poot et al., 2010*), alongside our implementation based on the sub-diffusion model (*Equation 3*), wherein mean kurtosis $K^*$ is computed as a function of the sub-diffusion model parameter $\beta$ (refer to (*Equation 9*)) using either a single or multiple diffusion times.

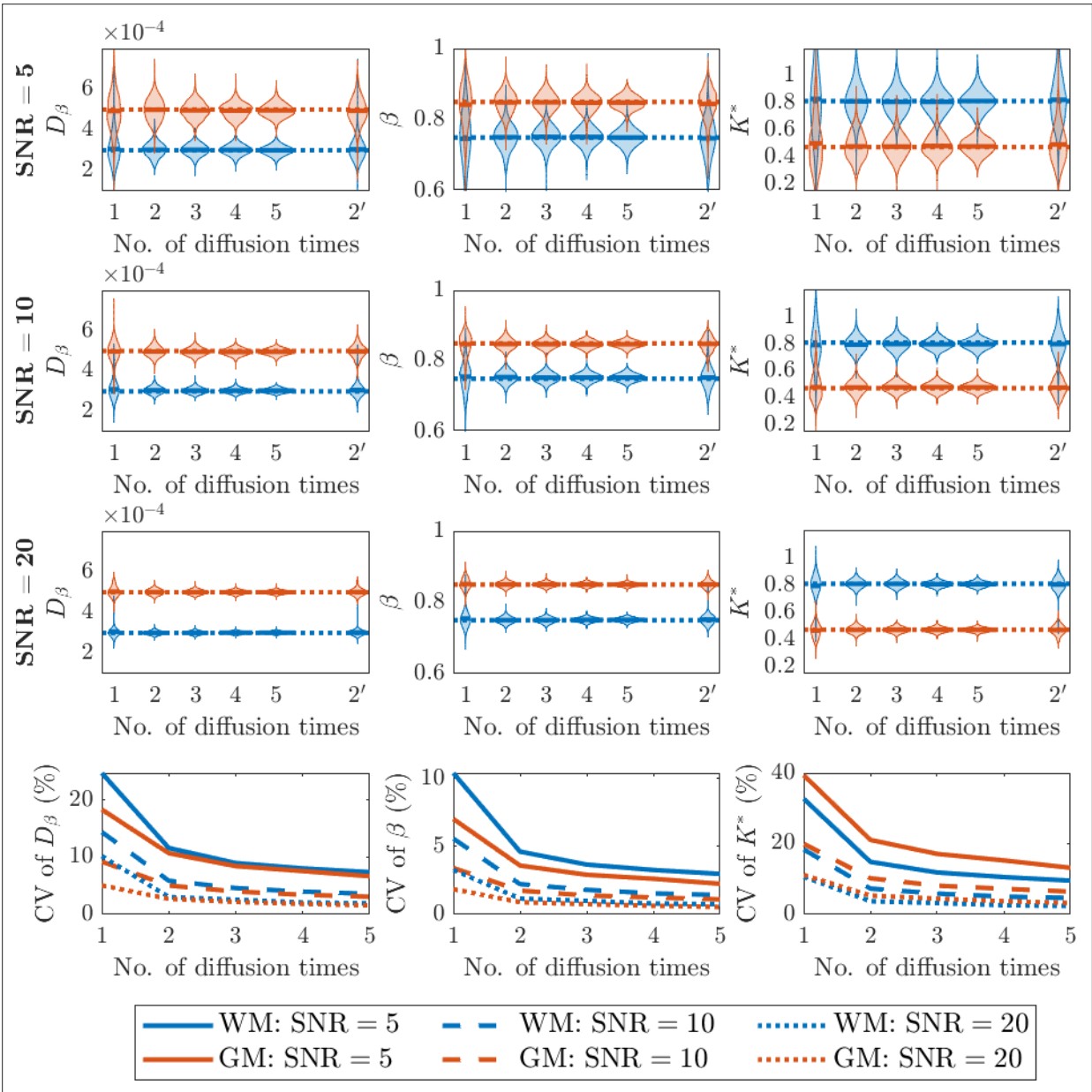

**Figure 1.** Simulation results on the effect of the number of diffusion times involved in the fitting of the sub-diffusion model (*Equation 3*) parameters ($D_\beta, \beta$) and computing $K^*$ following (*Equation 9*) at various SNR levels. The true values for ($D_\beta, \beta, K^*$) are set to (3 × 10⁻⁴, 0.75, 0.8125) to represent white matter (blue) and (5 × 10⁻⁴, 0.85, 0.4733) to represent grey matter (orange). Rows 1–3: Distributions of fitted parameter values using different number of diffusion times. 2′ represents an additional simulation using two diffusion times but set to be the same, so it has the same number of data points in the fitting as for using two different diffusion times. Row 4: Coefficient of variation (CV) of the parameter values fitted using different number of diffusion times.

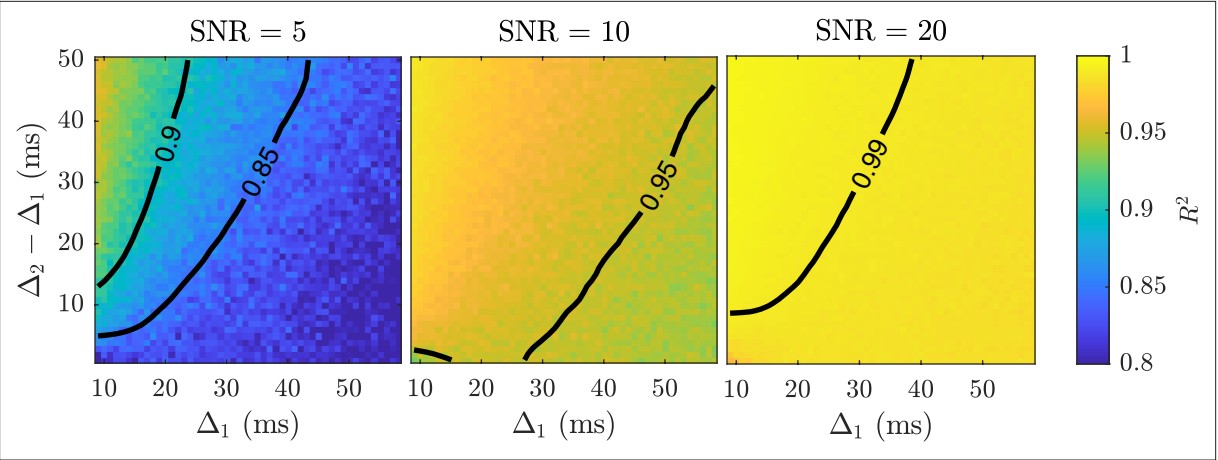

**Figure 2.** Surface plots of $R^2$ values achieved with fitting simulated data with two diffusion times, $\Delta_1$ and $\Delta_2$, to the sub-diffusion model (*Equation 3*) at various SNR levels. $R^2$ values were computed by comparing the estimated mean kurtosis with the true kurtosis. $R^2$ contours at the 0.85, 0.90, 0.95, and 0.99 levels have been provided for visualisation purposes.

## Multiple diffusion times for robust and accurate mean kurtosis estimation

In our simulations, we tested up to five distinct diffusion times to generate *b*-values. *Figure 1* illustrates the effects of the number of diffusion times on the parameter estimation at various SNR levels. We draw attention to a number of features in the plots. First, as SNR is increased from 5 to 20 (rows 1–3), the variability in the estimated parameters ($D_\beta$, $\beta$, $K^*$) decreases. Second, increasing the number of distinct diffusion times used for parameter estimation decreases estimation variability, with the most significant improvement when increasing from one to two diffusion times (rows 1–3). Third, sampling with two distinct diffusion times provides more robust parameter estimates than sampling twice as many *b*-values using one diffusion time (compare 2 and 2′ violin plots, rows 1–3). Fourth, the last row (row 4) highlights the improvement in the coefficient of variation (CV) for each parameter estimate with increasing SNR. This result again confirms that the most pronounced decline of CV occurs when increasing from one to two diffusion times, and parameter estimations can be performed more robustly using DW-MRI data with a relatively high SNR.

In *Figure 2*, we provide simulation results evaluating the choice of the two distinct diffusion times (assuming $\Delta_1 < \Delta_2$) by measuring the goodness-of-fit of the model. The smaller of the two diffusion times is stated along the abscissa, and the difference, i.e., $\Delta_2 - \Delta_1$, is plotted along the ordinate. The quality of fitting was measured using the coefficient of determination (the larger the $R^2$ value, the better the goodness-of-fit of the model) for each combination of abscissa and ordinate values. The conclusion from this figure is that $\Delta_1$ should be small, whilst $\Delta_2$ should be as large as practically plausible. Note, DW-MRI echo time was not considered in this simulation, but as $\Delta_2$ increases, the echo time has to proportionally increase. Because of the inherent consequence of decreasing SNR with echo time, special attention should be paid to the level of SNR achievable with the use of a specific $\Delta_2$. Nonetheless, our findings suggest that when $\Delta_1 = 8$ ms, $\Delta_2$ can be set as small as 21 ms to achieve an $R^2 > 0.90$ with an SNR as low as 5. If $\Delta_1$ is increased past 15 ms, then the separation between $\Delta_1$ and $\Delta_2$ has to increase as well, and such choices benefit from an increased SNR in the DW-MRI data. The Connectome 1.0 DW-MRI data was obtained using $\Delta_1 = 19$ ms and $\Delta_2 = 49$ ms, leading to a separation of 30 ms. For this data, it is expected that with SNR = 5 an $R^2$ around 0.9 is feasible, and by increasing SNR to 20, the $R^2$ can increase to a value above 0.99.

In *Figure 3*, we present the scatter plots of simulated *K* values versus fitted *K* values using simulated data with different number of diffusion times at various SNR levels. Four cases are provided, including fitting simulated data generated with $\Delta_1 = 19$ ms (row 1) or $\Delta_2 = 49$ ms (row 2), fitting data with both diffusion times (row 3), and fitting data with three diffusion times (row 4). $R^2$ values for each case at each SNR level are provided. This result verifies that sub-diffusion based kurtosis estimation (blue dots) improves using multiple diffusion times. The improvement in $R^2$ is prominent when moving from fitting single diffusion time data to two diffusion times data, especially when the data is noisy

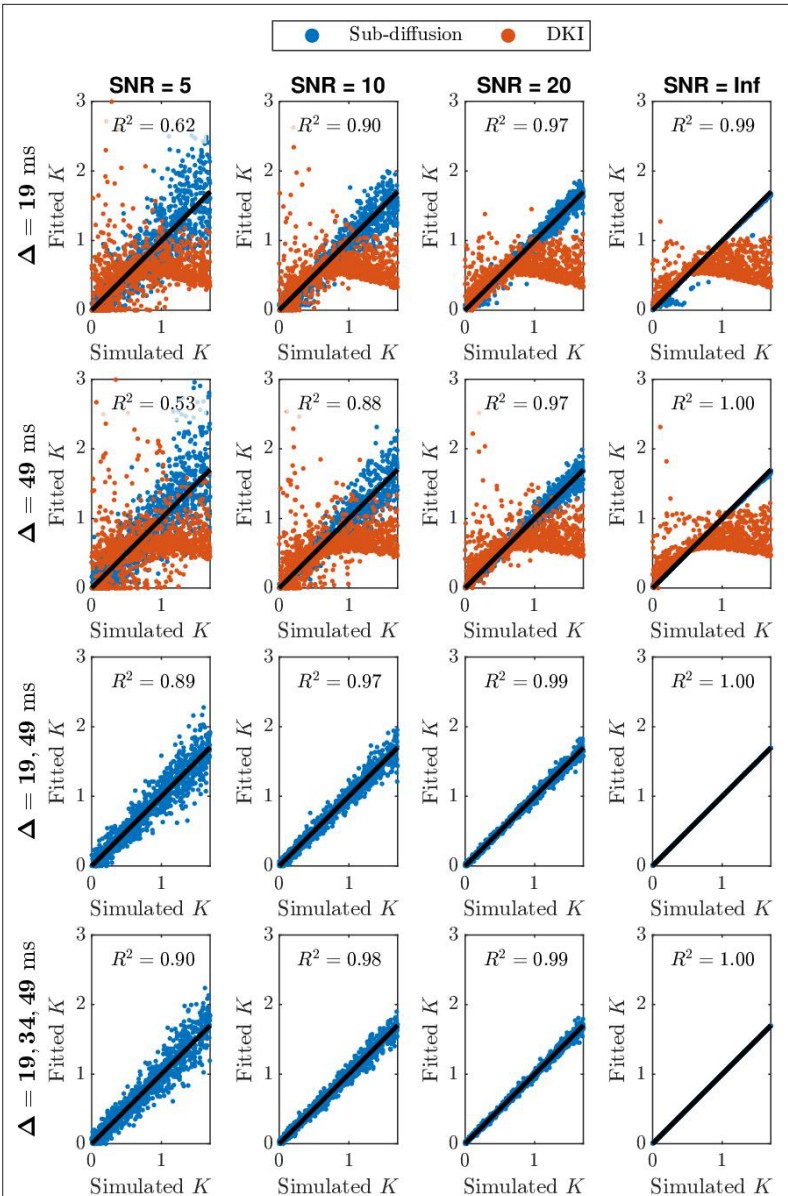

**Figure 3.** Scatter plots of simulated $K$ values versus fitted $K$ values for simulated data with different number of diffusion times at various SNR levels. The simulated data is created using the sub-diffusion model with random normal noise (*Equation 10*). Blue dots represent kurtosis based on fitting the sub-diffusion model (*Equation 3*). Orange dots represent kurtosis based on fitting the traditional diffusional kurtosis imaging (DKI) model (*Equation 6*). Black line is a reference line for $R^2 = 1.00$, indicating fitted kurtosis values are 100% matching the simulated ones.

(e.g., SNR = 5 and 10). The improvement gained by moving from two to three distinct diffusion times is marginal (less than 0.01 improvement in $R^2$ value at SNR = 5 and 10, and no improvements for SNR = 20 data). Moreover, our simulation findings highlight the deviation away from the true kurtosis $K$ by using the traditional DKI method (orange dots), especially with kurtosis values larger than 1. Overall, fitting sub-diffusion model (*Equation 3*) to data with two adequately separated diffusion times can lead to robust estimation of mean kurtosis, via (*Equation 9*).

## Time-dependence in DKI metrics

In *Figure 4*, we show the time-dependence effect of the DKI metrics ($D_{DKI}$ and $K_{DKI}$) after fitting the standard DKI model to our simulated data. In this fitting, we consider $b$-values of 0, 1000, 1400,

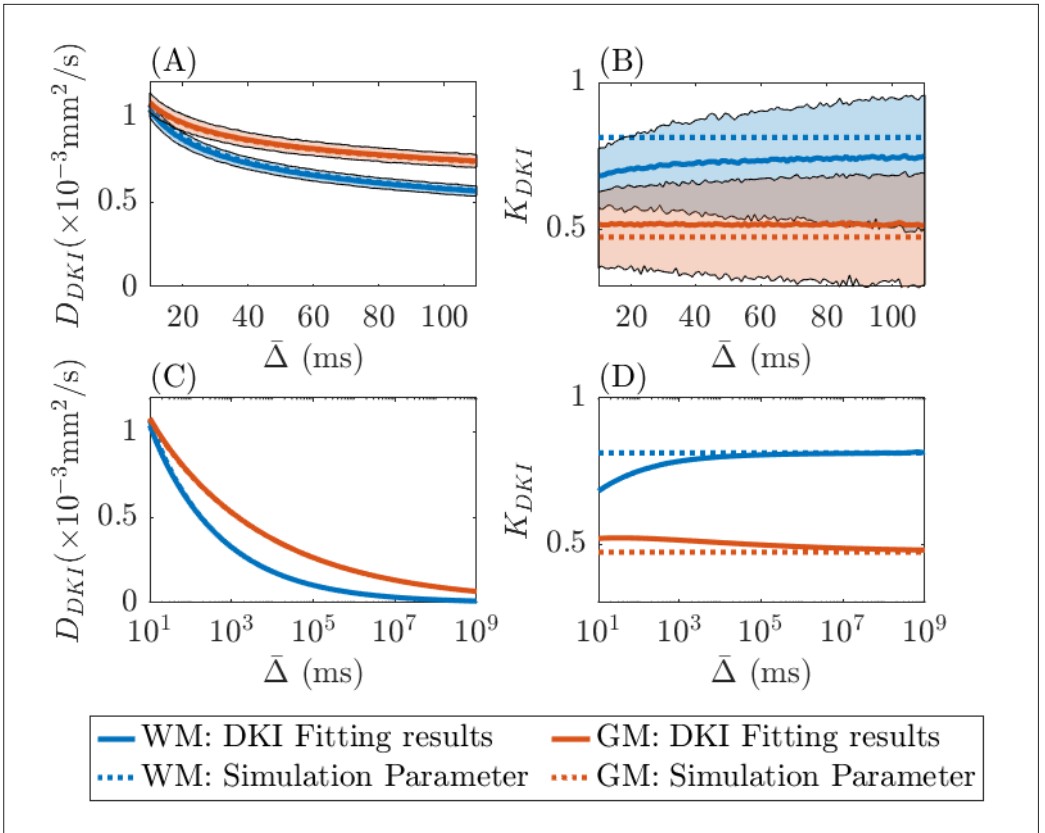

**Figure 4.** Time-dependence in diffusional kurtosis imaging (DKI) metrics using simulated data at different diffusion times ($\bar{\Delta}$). The true values for ($D_\beta$, $\beta$, $K^*$) used in the simulations are set to ($3 \times 10^{-4}$, 0.75, 0.8125) to represent white matter (blue dotted lines) and ($5 \times 10^{-4}$, 0.85, 0.4733) to represent grey matter (orange dotted lines). (**A**) and (**B**) use data with added random Gaussian noise (SNR = 20) to estimate the parameters $D_{DKI}$ and $K_{DKI}$. (**C**) and (**D**) use noiseless data to obtain estimates for large $\bar{\Delta}$ values. Shaded regions in (**A**) and (**B**) represent the 95% confidence intervals of the estimates.

and 2500 s/mm², and vary the diffusion time, as in *Jelescu et al., 2022*. We depict the parameter estimates, $D_{DKI}$ and $K_{DKI}$, from simulated data with added noise (SNR = 20) in (A) and (B) for the diffusion time between 10 and 110 ms. In (C) and (D), using data with no added noise, we illustrate the long-term fitting results and trends in the parameter estimates. In both (A) and (C), as diffusion time increases, $D_{DKI}$ decreases as expected for an effective diffusion coefficient. The mean $D_{DKI}$ (averaged over 1000 simulations) agrees with the true diffusivity $D^*$. When it comes to the kurtosis $K_{DKI}$, in the noiseless data setting (D), we see $K_{DKI}$ is converging to the true kurtosis value $K^*$ at large diffusion time, whilst in the noisy data setting (B), this trend is not obvious within the experimental diffusion time window. More explanations of the observed time-dependence of diffusivity and kurtosis are provided in the Discussion.

## Towards fast DKI data acquisitions

Next, we sought to identify the minimum number and combination of b-values to use for mean kurtosis estimation based on the sub-diffusion model (*Equation 3*). The simulation results were generated using the Connectome 1.0 DW-MRI data b-value setting, which has 16 b-values, 8 from $\Delta = 19$ ms and 8 from $\Delta = 49$ ms.

In *Figure 5*, we computed $R^2$ values for all combinations of choices when the number of non-zero b-values is 2, 3, and 4. We then plotted the $R^2$ values sorted in ascending order for each considered SNR level. Notably, as the number of non-zero b-values increase, the achievable combinations increase as well (i.e., 120, 560, and 1820 for the three different non-zero b-value cases). The colours illustrate the proportion of b-values used in the fitting based on $\Delta_1$ and $\Delta_2$. For example, 1:0 means only $\Delta_1$ b-values were used, and 2:1 means two $\Delta_1$ and one $\Delta_2$ b-values were used to generate the

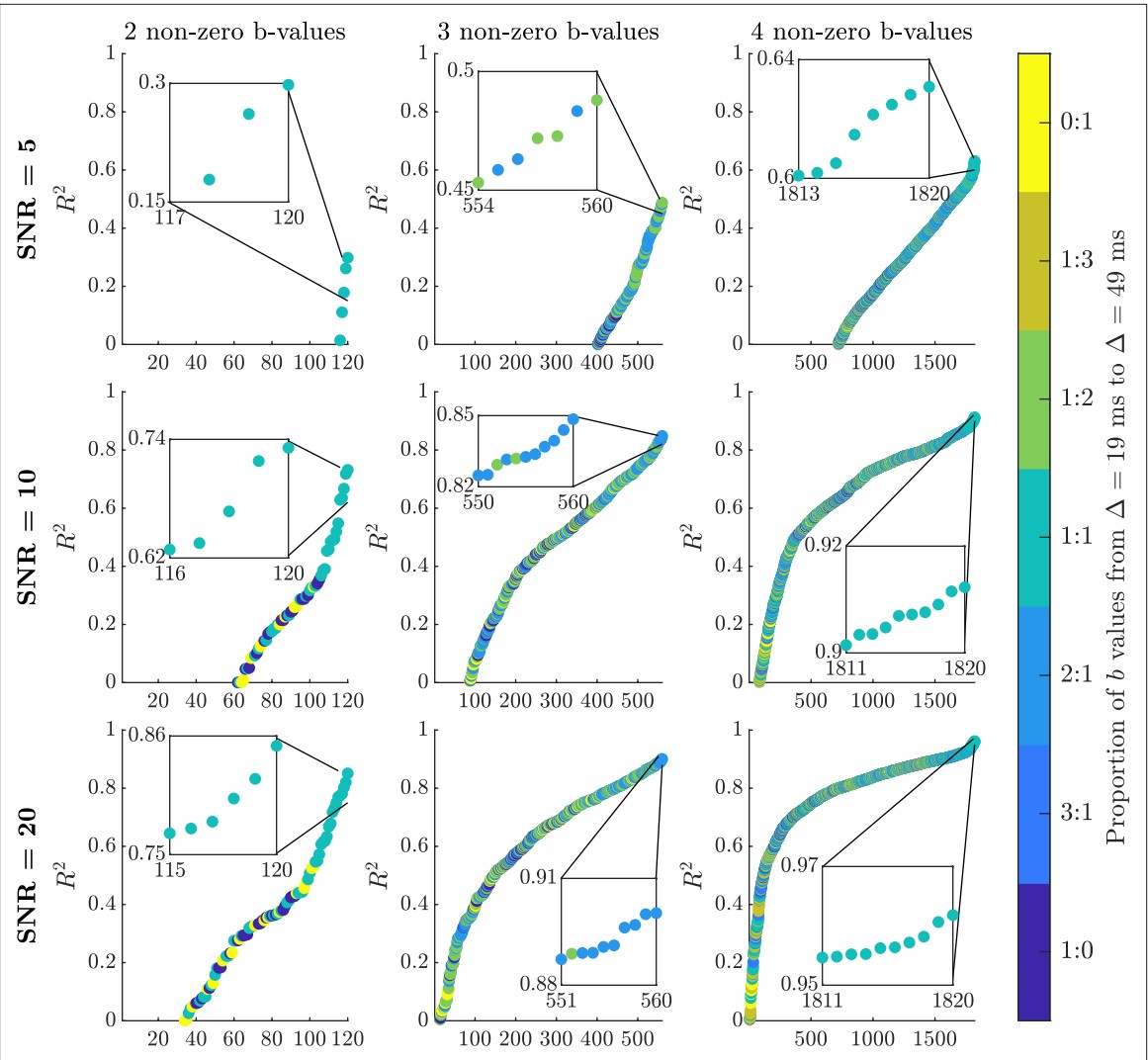

**Figure 5.** $R^2$ values for the *b*-value sampling optimisation based on DW-MRI data with SNR = 5, 10, and 20. The specifically investigated *b*-value combinations using two, three, and four non-zero *b*-values have been ordered by the size of the $R^2$ value. The colour bar depicts the proportion of $\Delta$ = 19 ms and $\Delta$ = 49 ms *b*-values needed to produce the corresponding $R^2$ value. Note, the different *b*-value combinations were assigned a unique identifier, and these appear along the abscissa for each of the three non-zero *b*-value cases. The *b*-value combinations achieving the highest $R^2$ values are displayed in the inset pictures and the *b*-values are provided in *Table 1*.

result. The *b*-value combinations achieving highest $R^2$ values are displayed in the inset pictures and the *b*-values are provided in *Table 1*.

Our simulation results in *Table 1* suggest that increasing the number of non-zero *b*-values from two to four improves the quality of the parameter estimation, also achievable by fitting the sub-diffusion model (*Equation 3*) to higher SNR data. The gain is larger by using higher SNR data than by using more *b*-values. For example, going from two to four non-zero *b*-values with SNR = 5 data approximately doubles the $R^2$, whereas the $R^2$ almost triples when SNR = 5 data is substituted by SNR = 20 data. Additionally, the use of $\Delta_1$ or $\Delta_2$ alone is not preferred (also see *Figure 5*), and preference is towards first using $\Delta_1$ and then supplementing with *b*-values generated using $\Delta_2$. At all SNR levels when only two non-zero *b*-values are used, one *b*-value should be chosen based on the $\Delta_1$ set, and the other based on $\Delta_2$. Moving to three non-zero *b*-values requires the addition of another $\Delta_1$ *b*-value, and when four non-zero *b*-values are used then two from each diffusion time are required. If we consider an $R^2$ = 0.90 to be a reasonable goodness-of-fit for the sub-diffusion model, then at least three or four non-zero *b*-values are needed with an SNR = 20. If SNR = 10, then three non-zero

**Table 1.** A selection of the best $b$-value sampling regimes to achieve the highest $R^2$ value in the three cases considered.

The various categories correspond with two, three and four non-zero $b$-value sampling schemes, with $\Delta_1$ and $\Delta_2$ denoting the diffusion time setting used to generate the $b$-values. Note, entries are $b$-values in unit of s/mm², and $\Delta_1 = 19$ ms and $\Delta_2 = 49$ ms were used to match the Connectome 1.0 DW-MRI data collection protocol. The entries listed at the bottom row are suggested optimal non-zero $b$-values for clinical practice.

| | Two $b$-values | | | Three $b$-values | | | Four $b$-values | | |
|---|---|---|---|---|---|---|---|---|---|
| | $\Delta_1$ | $\Delta_2$ | $R^2$ | $\Delta_1$ | $\Delta_2$ | $R^2$ | $\Delta_1$ | $\Delta_2$ | $R^2$ |
| | 350 | 6750 | 0.01 | 350, 800 | 2300 | 0.46 | 350, 4750 | 950, 4250 | 0.61 |
| | 800 | 2300 | 0.11 | 350 | 950, 4250 | 0.47 | 350, 6000 | 950, 6750 | 0.62 |
| SNR = 5 | 350 | 4250 | 0.18 | 350 | 2300, 4250 | 0.47 | 350, 4750 | 950, 6750 | 0.62 |
| | 350 | 950 | 0.26 | 50, 800 | 2300 | 0.48 | 350, 2400 | 950, 6750 | 0.63 |
| | 350 | 2300 | 0.30 | 350 | 950, 6750 | 0.49 | 350, 2400 | 950, 4250 | 0.63 |
| | 350 | 4250 | 0.63 | 350, 4750 | 2300 | 0.83 | 350, 2400 | 950, 9850 | 0.91 |
| | 800 | 4250 | 0.64 | 50, 800 | 2300 | 0.84 | 350, 3450 | 950, 6750 | 0.91 |
| SNR = 10 | 350 | 950 | 0.67 | 350, 800 | 2300 | 0.84 | 350, 6000 | 950, 9850 | 0.91 |
| | 350 | 2300 | 0.72 | 350, 1500 | 2300 | 0.84 | 350, 4750 | 950, 9850 | 0.91 |
| | 800 | 2300 | 0.73 | 350, 2400 | 2300 | 0.85 | 350, 3450 | 950, 9850 | 0.91 |
| | 1500 | 2300 | 0.77 | 350, 6000 | 2300 | 0.89 | 350, 4750 | 2300, 13,500 | 0.96 |
| | 350 | 950 | 0.78 | 50, 1500 | 4250 | 0.90 | 350, 4750 | 2300, 6750 | 0.96 |
| SNR = 20 | 350 | 2300 | 0.80 | 350, 1500 | 2300 | 0.90 | 350, 6000 | 2300, 9850 | 0.96 |
| | 800 | 4250 | 0.82 | 350, 4750 | 2300 | 0.90 | 350, 4750 | 2300, 9850 | 0.96 |
| | 800 | 2300 | 0.85 | 50, 2400 | 4250 | 0.90 | 350, 4750 | 950, 6750 | 0.96 |
| | 800 | 2300 | 0.85 | 350, 1500 | 2300 | 0.90 | 350, 1500 | 950, 4250 | 0.92 |

$b$-values will not suffice. Interestingly, an $R^2$ of 0.85 can still be achieved when SNR = 20 and two optimally chosen non-zero $b$-values are used.

*Table 1* summarises findings based on having different number of non-zero $b$-values with $R^2$ values deduced from the SNR = 5, 10, and 20 simulations. We have chosen to depict five $b$-value combinations producing the largest $R^2$ values for the two, three, and four non-zero $b$-value sampling cases. We found consistency in $b$-value combinations across SNR levels. Thus, we can conclude that a range of $b$-values can be used to achieve a large $R^2$ value, which is a positive finding, since parameter estimation does not stringently rely on $b$-value sampling. For example, using three non-zero $b$-values an $R^2 \geq 0.90$ is achievable based on different $b$-value sampling. Importantly, two distinct diffusion times are required, and preference is towards including a smaller diffusion time $b$-value first. Hence, for three non-zero $b$-values we find two $b$-values based on $\Delta_1$ and one based on $\Delta_2$. This finding suggests one of the $\Delta_1$ $b$-values can be chosen in the range 50 to 350 s/mm², and the other in the range 1500 to 4750 s/mm². Additionally, the $\Delta_2$ $b$-value can also be chosen in a range, considering between 2300 to 4250 s/mm² based on the Connectome 1.0 $b$-value settings. The $b$-value sampling choices made should nonetheless be in view of the required $R^2$ value. Overall, sampling using two distinct diffusion times appears to provide quite a range of options for the DW-MRI data to be used to fit the sub-diffusion model parameters. The suggested optimal $b$-value sampling in the last row of *Table 1*, primarily chosen to minimise $b$-value size whilst maintaining a large $R^2$ value, may be of use for specific neuroimaging studies, which will be used to inform our discussion on feasibility for clinical practice.

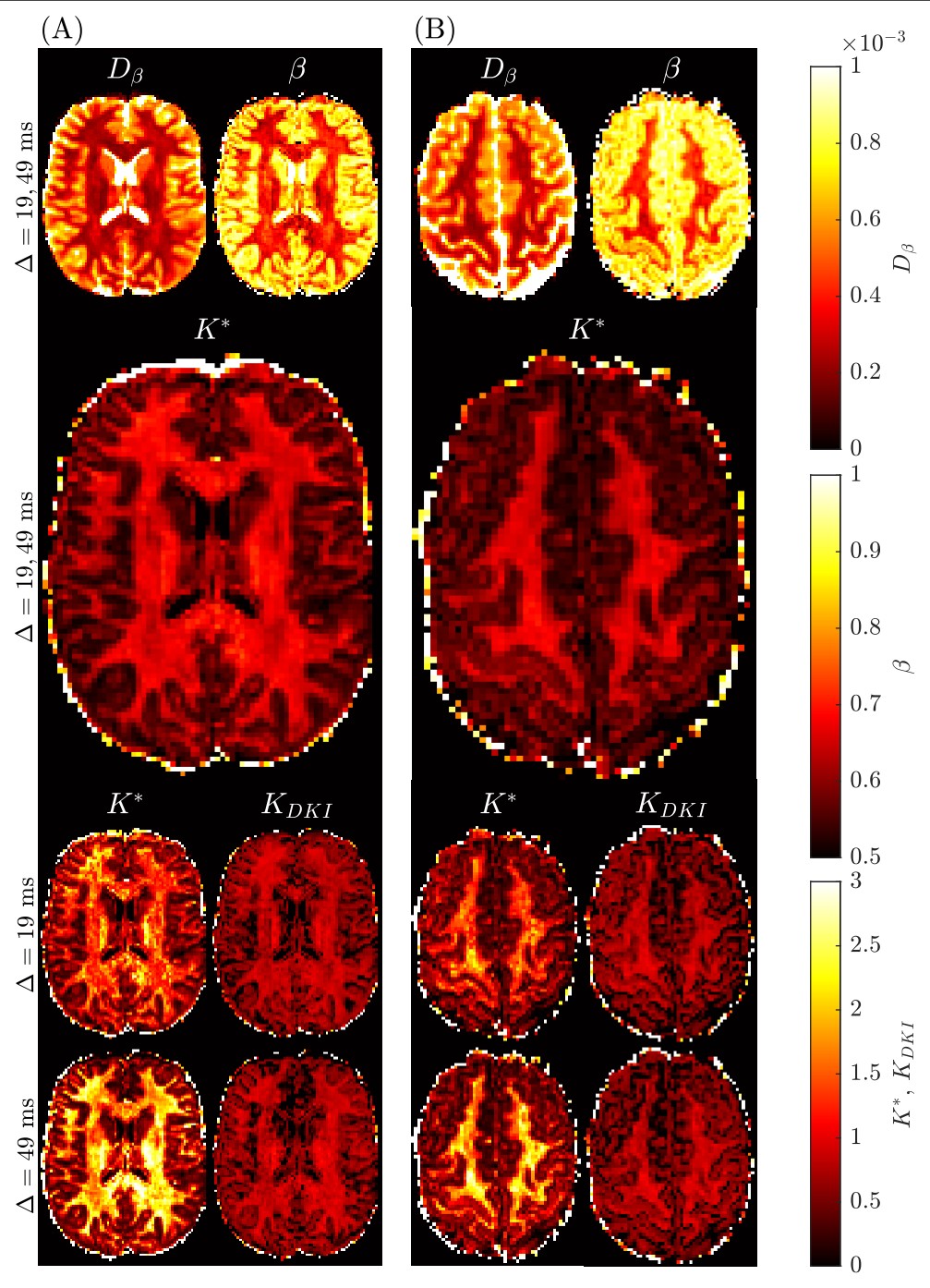

**Figure 6.** Spatially resolved maps of mean kurtosis shown for two example slices and two different subjects, Subject 3 rescan slice 71 (Panel **A**) and Subject 5 slice 74 (Panel **B**) from the Connectome 1.0 DW-MRI data. Individual maps were generated using the sub-diffusion model framework ($K^*$), as well as using the traditional approach ($K_{DKI}$). The diffusion times, $\Delta$, used to generate each plot are provided for each case. We consider the mean kurtosis maps using two diffusion times ($\Delta = 19, 49$ ms) as the benchmarks.

## Benchmark mean kurtosis in the brain

The benchmark mean kurtosis estimation in the brain is established using the entire *b*-value range with all diffusion encoding directions available in the Connectome 1.0 DW-MRI data. For two subjects in different slices, *Figure 6* provides the spatially resolved maps of mean kurtosis computed using the sub-diffusion method (i.e., $K^*$) with one or two diffusion times, and using the standard method (i.e.,

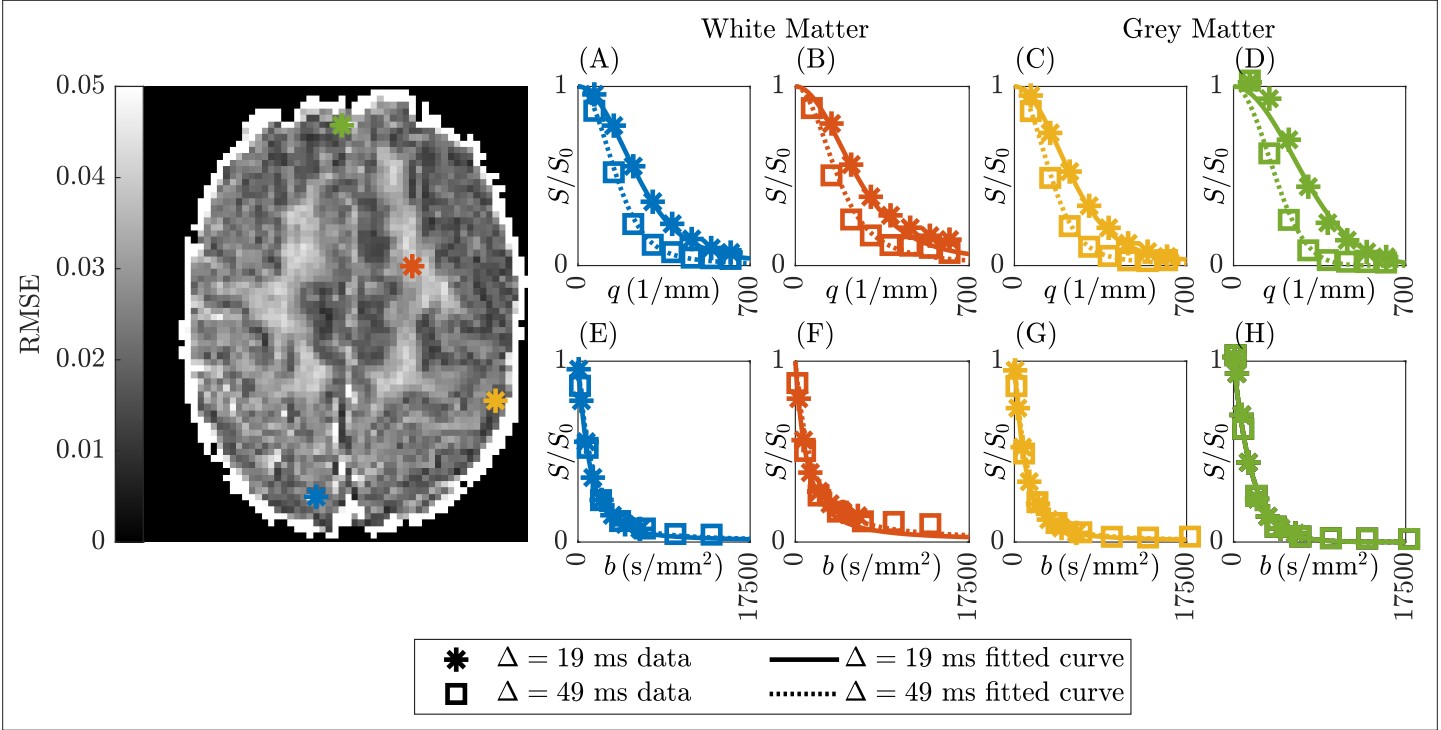

**Figure 7.** Representative error map and sample parameter fits for Subject 5 slice 74. The DW-MRI data with two diffusion times was fitted to the sub-diffusion model in both $q$-space (**A–D**) and $b$-space (**E–H**), following (*Equation 3*) and (*Equation 4*), respectively. The first and second columns are voxels in white matter (30,20,74) and (45,56,74), respectively. The third and fourth columns are voxels in grey matter (58,35,74) and (34,78,74), respectively.

$K_{DKI}$) considering the two distinct diffusion times. First, we notice a degradation in the $K_{DKI}$ image with an increase in diffusion time. Second, the use of a single diffusion time with the sub-diffusion model leads to $K^*$ values which are larger than either the $K_{DKI}$ values or $K^*$ values generated using two diffusion times. Third, the quality of the mean kurtosis map appears to visually be best when two diffusion times are used to estimate $K^*$. Superior grey-white matter tissue contrast (TC) was found for the $K^*$ map (TC = 1.73), compared to the $K_{DKI}$ maps (TC = 0.80 for the $\Delta = 19$ ms dataset and TC = 1.01 for the $\Delta = 49$ ms dataset).

In *Figure 7*, an error map (measured by root-mean-square-error, RMSE) from Subject 5 slice 74 was presented for fitting the sub-diffusion model to the DW-MRI data with two diffusion times. Sample parameter fittings in both $b$-space (*Equation 3*) and $q$-space (*Equation 4*) were provided for four representative white and grey matter voxels.

Quantitative findings for kurtosis are provided in *Table 2*. The analysis was performed for sub-cortical grey matter (scGM), cortical grey mater (cGM) and white matter (WM) brain regions. For specifics we refer the reader to the appropriate methods section. The table entries highlight the differences in mean kurtosis when computed using the two different approaches. The trend for the traditional single diffusion time approach is that an increase in $\Delta$ results in a slight decrease in the mean $K_{DKI}$, and an increase in the coefficient of variation (CV) for any region. For example, the mean $K_{DKI}$ in the thalamus reduces from 0.65 to 0.58, whilst the CV increases from 30% to 39%. As much as 30% increase in CV is common for scGM and cGM regions, and around 10% for WM regions. The CV based on the $K^*$ value for each region is less than the CV for $K_{DKI}$ with either $\Delta = 19$ ms or $\Delta = 49$ ms.

*Figure 8* presents the distributions of the fitted parameters ($D$ and $K$) in specific brain regions, based on the sub-diffusion model (Panel A) and the standard DKI model with $\Delta = 19$ ms (Panel B). The distributions are coloured by the probability density. Yellow indicates high probability density, light blue indicates low probability density. In each subplot, the diffusivity is plotted along the abscissa axis and the kurtosis is along the ordinate axis. Results for the standard DKI model with $\Delta = 49$ ms are qualitatively similar, so are not shown here. From Panel (B), we see an unknown nonlinear relationship between the DKI pair, $D_{DKI}$ and $K_{DKI}$, in all regions considered. By comparison, the sub-diffusion based $K^*$ and $D^*$ (Panel A) are less correlated with each other, indicating $D^*$

**Table 2.** Benchmark kurtosis values estimated using the Connectome 1.0 DW-MRI data for different regions of the human brain.

Results are provided for the traditional mean kurtosis ($K_{DKI}$) at two distinct diffusion times, and values ($K^*$) obtained based on fitting the sub-diffusion model across both diffusion times. Results are for grey matter (GM) and white matter (WM) brain regions, in categories of sub-cortical (sc) and cortical (c), and CC stands for corpus callosum. The pooled means and standard deviations across participants have been tabulated, along with the coefficient of variation in parentheses.

| | $K_{DKI}$ | | $K^*$ |
| | $\Delta = 19$ ms | $\Delta = 49$ ms | $\Delta = 19, 49$ ms |
|---|---|---|---|
| **scGM** | 0.57 ± 0.23(40%) | 0.50 ± 0.24(48%) | 0.60 ± 0.21(35%) |
| Thalamus | 0.65 ± 0.19(30%) | 0.58 ± 0.22(39%) | 0.70 ± 0.17(25%) |
| Caudate | 0.41 ± 0.24(58%) | 0.37 ± 0.23(64%) | 0.39 ± 0.14(35%) |
| Putamen | 0.54 ± 0.21(40%) | 0.45 ± 0.22(49%) | 0.49 ± 0.13(27%) |
| Pallidum | 0.68 ± 0.25(37%) | 0.64 ± 0.26(41%) | 0.93 ± 0.18(19%) |
| **cGM** | 0.53 ± 0.24(46%) | 0.46 ± 0.23(51%) | 0.40 ± 0.16(39%) |
| Fusiform | 0.55 ± 0.22(40%) | 0.44 ± 0.22(49%) | 0.40 ± 0.15(37%) |
| Lingual | 0.59 ± 0.21(35%) | 0.53 ± 0.22(42%) | 0.47 ± 0.16(34%) |
| **WM** | 0.78 ± 0.20(25%) | 0.76 ± 0.19(26%) | 0.87 ± 0.22(25%) |
| Cerebral WM | 0.77 ± 0.19(25%) | 0.75 ± 0.19(26%) | 0.85 ± 0.22(26%) |
| Cerebellum WM | 0.99 ± 0.18(18%) | 0.95 ± 0.19(20%) | 1.07 ± 0.22(21%) |
| CC | 0.65 ± 0.22(35%) | 0.65 ± 0.22(34%) | 0.95 ± 0.25(26%) |

and $K^*$ carry distinct information, which will be very valuable for characterising tissue microstructure. **Tables 3** and **Table 4** present the $D_\beta$ and $\beta$ values used to compute $D^*$ and $K^*$ in **Table 2** and **Figure 8**.

## Reduction in DKI data acquisition

Results for reductions in diffusion encoding directions to achieve different levels of SNR with the purpose of shortening acquisition times will be benchmarked against the $K^*$ maps in **Figure 6** and the $K^*$ values reported in **Table 2**.

**Figure 9** presents the qualitative findings for two subjects generated using all, optimal and sub-optimal $b$-value samplings with SNR = 6 (3 non-collinear directions, 6 measurements), 10 (8 non-collinear directions, 16 measurements), and 20 (32 non-collinear directions, 64 measurements) DW-MRI data. The quality of the mean kurtosis map improves with increasing SNR, and also by optimising $b$-value sampling. Optimal sampling at SNR = 10 is qualitatively comparable to the SNR = 20 optimal sampling result, and to the benchmark sub-diffusion results in **Figure 6**.

Quantitative verification of the qualitative observations are provided in **Table 5**. Significant differences in brain region-specific mean kurtosis values occur at the SNR = 6 level, which are not apparent when SNR = 10 or 20 data with optimal $b$-value sampling were used. The average errors are relative errors compared to the benchmark kurtosis values reported in **Table 2**. When using optimal $b$-values, average errors range from 22% to 43% at SNR = 6, from 13% to 43% at SNR = 10, and from 8% to 20% at SNR = 20, across brain regions. When using sub-optimal $b$-values, average errors range from 47% to 57% at SNR = 6, from 24% to 102% at SNR = 10, and from 27% to 72% at SNR = 20. Also, the brain region-specific CV for mean kurtosis was not found to change markedly when SNR = 10 or 20 data were used to compute $K^*$. The result of reducing the SNR to 6 leads to an approximate doubling of the CV for each brain region. These findings confirm that with optimal $b$-value sampling, the data quality can be reduced to around the SNR = 10 level, without a significant impact on the region-specific mean kurtosis estimates derived using the sub-diffusion model.

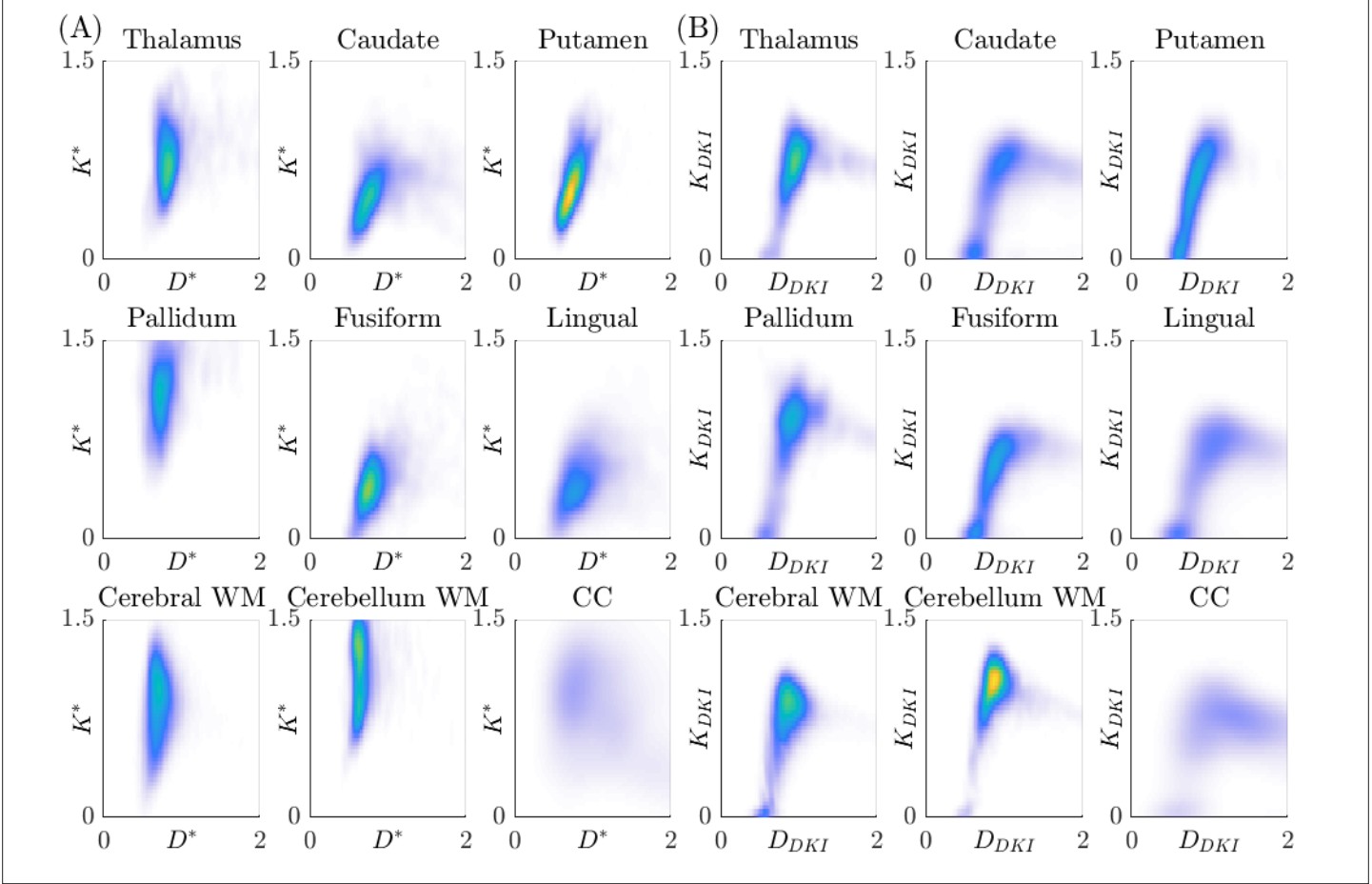

**Figure 8.** Distributions of the estimated parameter pair $(D, K)$ in different regions of the brain of all subjects, coloured by the probability density. Yellow indicates high probability density, light blue indicates low probability density. Panel (**A**): distributions of $(D^*, K^*)$, generated using the sub-diffusion model (*Equation 3*) with both $\Delta = 19, 49$ ms. Panel (**B**): distributions of $(D_{DKI}, K_{DKI})$, generated using the standard diffusional kurtosis imaging (DKI) model (*Equation 6*) with $\Delta = 19$ ms. Kurtosis is dimensionless and diffusivity is in units of $\times 10^{-3}$ mm²/s.

## Scan–rescan reproducibility of mean kurtosis

*Figure 10* summarises the intraclass correlation coefficient (ICC) distribution results ($\mu$ for mean; $\sigma$ for standard deviation) for the specific brain regions analysed. The two sets of ICC values were computed based on all DW-MRI (i.e., SNR = 20; subscript A) and the SNR = 10 optimal *b*-value sampling scheme (subscript O). As the value of $\mu$ approaches 1, the inter-subject variation in mean kurtosis is expected to greatly outweigh the intra-subject scan–rescan error. The value of $\mu$ should always be above 0.5, otherwise parameter estimation cannot be performed robustly and accurately, and values above 0.75 are generally accepted as good. The $\mu_A$ values for all regions were in the range 0.76 (thalamus) to 0.87 (caudate), and reduced to the range 0.57 (thalamus) to 0.80 (CC) when optimal sampling with SNR = 10 was used to estimate the $K^*$ value. Irrespective of which of the two DW-MRI data were used for $K^*$ estimation, the value of $\mu$ was greater than or equal to 0.70 in 20 out of 24 cases. The $\mu_O$ was less than 0.70 for only the thalamus, putamen, and pallidum. The loss in ICC by going to SNR = 10 data with optimal *b*-value sampling went hand-in-hand with an increase in $\sigma$, which is not unexpected, since the uncertainty associated with using less data should be measurable. Overall, $\mu_A$, $\mu_O$, and $\sigma_A$, $\sigma_O$, were fairly consistent across the brain regions, suggesting the DW-MRI data with SNR = 10 is sufficient for mean kurtosis estimation based on the sub-diffusion framework.

**Table 3.** Benchmark $D_\beta$ values estimated using the Connectome 1.0 DW-MRI data for different regions of the human brain.

Results are provided for fitting the data at two distinct diffusion times, and fitting the sub-diffusion model across both diffusion times. Results are for grey matter (GM) and white matter (WM) brain regions, in categories of sub-cortical (sc) and cortical (c), and CC stands for corpus callosum. The pooled means and standard deviations across participants have been tabulated, along with the coefficient of variation in parentheses.

| | $D_\beta(\times 10^{-4}\,\mathrm{mm^2/s^\beta})$ | | |
| --- | --- | --- | --- |
| | $\Delta = 19$ ms | $\Delta = 49$ ms | $\Delta = 19, 49$ ms |
| **scGM** | 3.12 ± 0.96(31%) | 3.56 ± 1.03(29%) | 4.13 ± 0.86(21%) |
| Thalamus | 2.73 ± 0.83(30%) | 3.14 ± 0.89(28%) | 3.89 ± 0.81(21%) |
| Caudate | 4.32 ± 0.94(22%) | 4.80 ± 0.89(19%) | 5.21 ± 1.03(20%) |
| Putamen | 3.36 ± 0.58(17%) | 3.96 ± 0.54(14%) | 4.31 ± 0.44(10%) |
| Pallidum | 1.84 ± 0.63(34%) | 1.88 ± 0.67(36%) | 2.79 ± 0.52(19%) |
| **cGM** | 4.17 ± 0.97(23%) | 5.19 ± 0.96(19%) | 5.32 ± 0.90(17%) |
| Fusiform | 3.94 ± 0.83(21%) | 4.93 ± 0.74(15%) | 5.05 ± 0.69(14%) |
| Lingual | 3.78 ± 1.00(27%) | 4.62 ± 0.99(21%) | 5.01 ± 1.02(20%) |
| **WM** | 1.84 ± 0.79(43%) | 2.14 ± 0.93(43%) | 2.94 ± 0.72(24%) |
| Cerebral WM | 1.90 ± 0.77(41%) | 2.20 ± 0.92(42%) | 2.98 ± 0.71(24%) |
| Cerebellum WM | 0.81 ± 0.44(55%) | 1.17 ± 0.61(52%) | 2.03 ± 0.49(24%) |
| CC | 2.58 ± 1.84(71%) | 2.49 ± 1.91(76%) | 3.78 ± 2.12(56%) |

## Discussion

DW-MRI allows the measurement of mean kurtosis, a metric for the deviation away from standard Brownian motion of water in tissue, which has been used to infer variations in tissue microstructure. Research on mean kurtosis has shown benefits in specific applications over other diffusion related measures derived from DW-MRI data (*Li et al., 2022b*; *Liu et al., 2022*; *Huang et al., 2022*; *Guo et al., 2022b*; *Goryawala et al., 2023*; *Guo et al., 2022a*; *Wang et al., 2022*; *Li et al., 2022a*; *Maral-akunte et al., 2023*; *Hu et al., 2022*; *Zhou et al., 2023*; *Li et al., 2022c*). Whilst many efforts have been made to optimise mean kurtosis imaging for clinical use, the limitations have been associated with lack of robustness and the time needed to acquire the DW-MRI data for mean kurtosis estimation. The choice of the biophysical model and how diffusion encoding is applied are critical to how well kurtosis in the brain is mapped. Here, we evaluated the mapping of mean kurtosis based on the sub-diffusion model, which allows different diffusion times to be incorporated into the data acquisition. Using simulations and the Connectome 1.0 public DW-MRI dataset, involving a range of diffusion encodings, we showed that mean kurtosis can be mapped robustly and rapidly provided at least two different diffusion times are used and care is taken towards how *b*-values are chosen given differences in the SNR level of different DW-MRI acquisitions.

### Reduction in scan time

Previous attempts have been made in optimising the DW-MRI acquisition protocol for mean kurtosis estimation based on the traditional, single diffusion time kurtosis model (*Hansen et al., 2013*; *Hu et al., 2022*; *Poot et al., 2010*; *Hansen et al., 2016*). Considerations have been made towards reducing the number of b-shells, directions per shell, and sub-sampling of DW-MRI for each direction in each shell. Our findings suggest that robust estimation of kurtosis cannot be achieved using the classical model for mean kurtosis, as highlighted previously (*Yang et al., 2022*; *Ingo et al., 2014*; *Ingo et al., 2015*; *Barrick et al., 2020*). A primary limitation of the traditional method is the use of the cumulant expansion resulting in sampling below a *b*-value of around 2500 s/mm$^2$ (*Jensen et al., 2005*;

**Table 4.** Benchmark $\beta$ values estimated using the Connectome 1.0 DW-MRI data for different regions of the human brain.

Results are provided for fitting the data at two distinct diffusion times, and fitting the sub-diffusion model across both diffusion times. Results are for grey matter (GM) and white matter (WM) brain regions, in categories of sub-cortical (sc) and cortical (c), and CC stands for corpus callosum. The pooled means and standard deviations across participants have been tabulated, along with the coefficient of variation in parentheses.

| | $\beta$ | | |
|---|---|---|---|
| | $\Delta = 19$ ms | $\Delta = 49$ ms | $\Delta = 19, 49$ ms |
| **scGM** | 0.74 ± 0.10(13%) | 0.69 ± 0.14(19%) | 0.81 ± 0.06(8%) |
| Thalamus | 0.70 ± 0.09(13%) | 0.63 ± 0.12(20%) | 0.78 ± 0.05(6%) |
| Caudate | 0.83 ± 0.07(8%) | 0.81 ± 0.07(9%) | 0.88 ± 0.04(5%) |
| Putamen | 0.79 ± 0.07(8%) | 0.77 ± 0.06(8%) | 0.85 ± 0.04(5%) |
| Pallidum | 0.61 ± 0.11(19%) | 0.47 ± 0.14(30%) | 0.72 ± 0.05(7%) |
| **cGM** | 0.81 ± 0.08(10%) | 0.83 ± 0.07(9%) | 0.87 ± 0.05(5%) |
| Fusiform | 0.81 ± 0.08(9%) | 0.83 ± 0.07(8%) | 0.87 ± 0.04(5%) |
| Lingual | 0.78 ± 0.09(11%) | 0.78 ± 0.09(12%) | 0.85 ± 0.05(6%) |
| **WM** | 0.61 ± 0.12(20%) | 0.52 ± 0.16(32%) | 0.74 ± 0.06(9%) |
| Cerebral WM | 0.62 ± 0.12(19%) | 0.53 ± 0.16(30%) | 0.74 ± 0.06(8%) |
| Cerebellum WM | 0.41 ± 0.16(39%) | 0.32 ± 0.19(60%) | 0.68 ± 0.06(9%) |
| CC | 0.61 ± 0.14(23%) | 0.43 ± 0.18(43%) | 0.71 ± 0.07(10%) |

*Jensen and Helpern, 2010*), and using the sub-diffusion model this limitation is removed (*Yang et al., 2022*). Our simulation findings and experiments using the Connectome 1.0 data confirm that mean kurtosis can be mapped robustly and rapidly using the sub-diffusion model applied to an optimised DW-MRI protocol. Optimisation of the data acquisition with the use of the sub-diffusion model has not been considered previously.

Mean kurtosis values can be generated based on having limited number of diffusion encoding directions (refer to *Figure 9* and *Table 5*). Given that each direction for each b-shell takes a fixed amount of time, then a four b-shell acquisition with six directions per shell will take 25 times longer than a single diffusion encoding data acquisition (assuming a single *b*-value=0 data is collected). The total acquisition time for the diffusion MRI protocol for the Connectome 1.0 data was 55 min, including 50 $b = 0$ s/mm$^2$ scans plus seven *b*-values with 32 and nine with 64 diffusion encoding directions (*Tian et al., 2022*). This gives a total of 850 scans per dataset. As such, a single 3D image volume took 3.88 s to acquire. Conservatively allowing 4 s per scan, and considering SNR = 20 data (i.e., 64 directions) over four *b*-values and a single *b*-value = 0 scan, DW-MRI data for mean kurtosis estimation can be completed in 17 min 8 s ($R^2 = 0.96$). At SNR = 10 (i.e., 32 directions), DW-MRI data with the same number of *b*-values can be acquired in 4 min 20 s ($R^2 = 0.91$). If an $R^2 = 0.85$ (SNR = 10) is deemed adequate, then one less b-shell is required, saving an additional 64 s. We should point out that even though 2-shell optimised protocols can achieve $R^2 = 0.85$ with SNR = 20, this is not equivalent in time to using 3-shells with SNR = 10 (also $R^2 = 0.85$). This is because 4×4× additional data are required to double the SNR (equivalent to acquiring an additional 4-shells). However, only one extra b-shell is required to convert 2-shell data to 3-shells with SNR = 10. Our expected DW-MRI data acquisition times are highly feasible clinically, where generally neuroimaging scans take around 15 min involving numerous different MRI contrasts and often a DTI acquisition.

An early study on estimating mean kurtosis demonstrated the mapping of a related metric in less than 1 min over the brain (*Hansen et al., 2013*). Clinical adoption of the protocol lacked, possibly since b-values are a function $\delta$, $G$, and $\Delta$. Hence, different *b*-values can be obtained using different DW-MRI protocol settings, leading to differences in the mapping of mean kurtosis based on the data

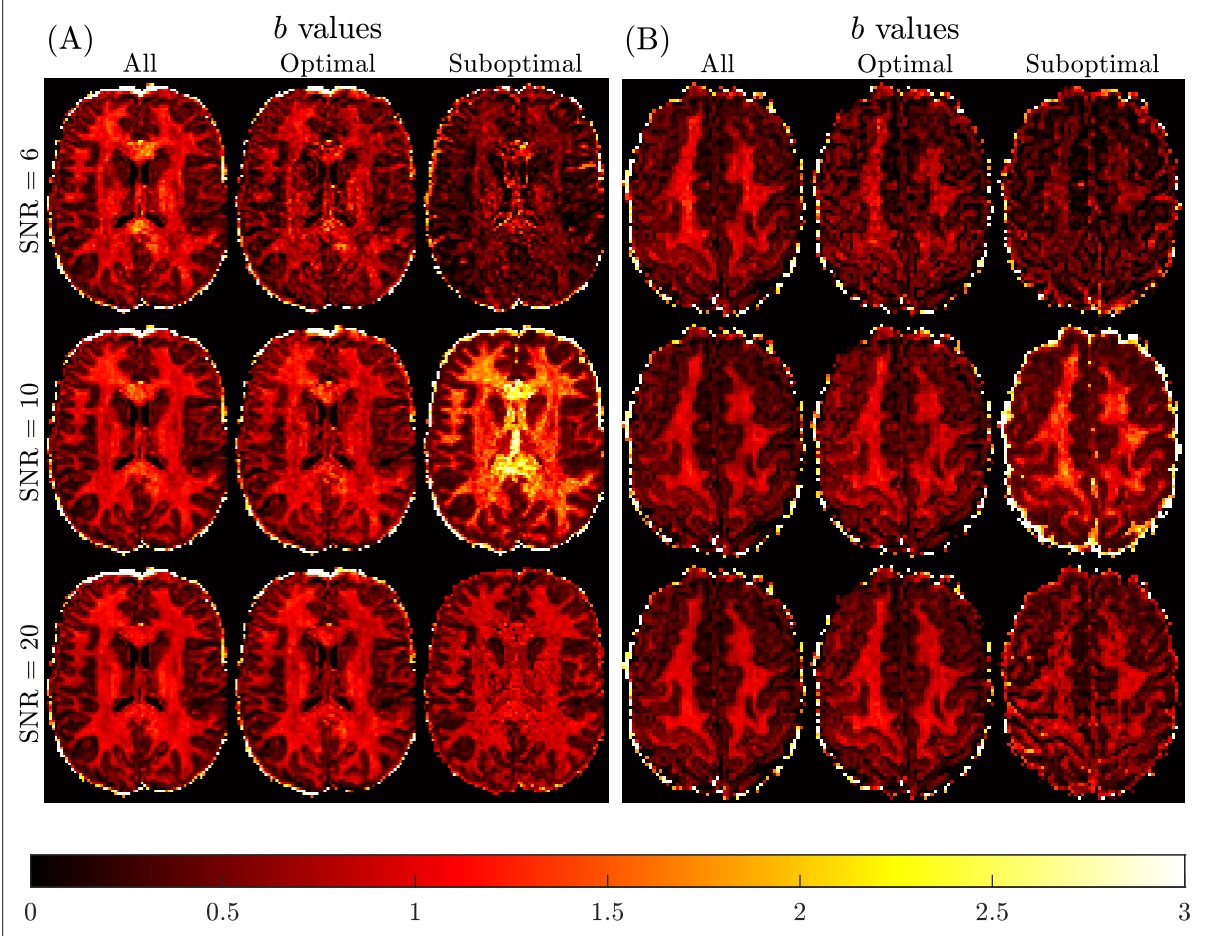

**Figure 9.** Spatially resolved maps of mean kurtosis shown for two example slices and two different subjects, Subject 3 rescan slice 71 (Panel **A**) and Subject 5 slice 74 (Panel **B**), based on SNR reduction of the Connectome 1.0 DW-MRI data. Individual maps were generated using the sub-diffusion model framework ($K^*$), considering optimal and sub-optimal four non-zero $b$-value sampling schemes. Here, two $b$-values with $\Delta = 19$ ms and two $b$-values with $\Delta = 49$ ms were selected for each case. The optimal $b$-values were chosen as the best for each SNR shown in **Table 1**. The sub-optimal $b$-values were chosen to have an $R^2$ = 0.3, 0.45, 0.5 to be about half of the maximum $R^2$, for SNR = 6 ($b = 800, 1500, 200, 2300$ s/mm$^2$), SNR=10 ($b = 1500, 3450, 6750, 13500$ s/mm$^2$), and SNR = 20 ($b = 3450, 4750, 2300, 4250$ s/mm$^2$), respectively. The benchmark kurtosis map is provided in **Figure 6**.

(we showed the $\Delta$ effect in **Figure 6** and **Table 2**). Our findings suggest this impediment is removed by sampling and fitting data with $b$-values across two distinct diffusion times. Nonetheless, we should consider what might be an acceptable DW-MRI data acquisition time.

A recent study on nasopharyngeal carcinoma investigated reducing the number of b-shell signals based on fixing diffusion encoding directions to the three Cartesian orientations (**He et al., 2022**). The 3-shell acquisitions took $3 \min 2$ s to acquire, whilst the 5-shell data required 5 min 13 s. They investigated as well the impact of using partial Fourier sampling, i.e., reducing the amount of data needed for image reconstruction by reducing $k$-space line acquisitions for each diffusion encoded image. Their benchmark used 5-shells (200, 400, 800, 1500, 2000 s/mm$^2$), and found partial Fourier sampling with omission of the 1500 s/mm$^2$ b-shell produced acceptable results. With this acquisition the scan could be completed in 3 min 46 s, more than 2 times faster than the benchmark 8 min 31 s. Our proposed 3-shells acquisition with an $R^2$ = 0.85 (see SNR = 10 results in **Table 1**) executable under 4 min is therefore inline with current expectations. Note, at the $R^2$ = 0.85 level the ICC for the different brain regions were in the range 0.60–0.69, and these were not formally reported in **Figure 10**. This level of reproducibility is still acceptable for routine use. We should additionally point out that we used the Subject 1 segmentation labels, after having registered each DW-MRI data to the Subject 1 first scan. This approach results in slight mismatch of the region-specific segmentations when carried across subjects, inherently resulting in an underestimation of ICC values.

**Table 5.** Kurtosis values ($K^*$) under the optimal and sub-optimal $b$-value sampling regimes for specific brain regions. $K^*$ was estimated based on fitting the sub-diffusion model to the Connectome 1.0 DW-MRI data with two diffusion times and selected four b-shells. Optimal $b$-value sampling is considered to have $R^2$ = 0.63, 0.91 and 0.96 for the SNR = 6, 10, and 20 columns, according to *Table 1*. Sub-optimal $b$-values are chosen to have $R^2$ = 0.3, 0.45, and 0.5, respectively, as reported in *Figure 9*. Individual entries are for grey matter (GM) and white matter (WM) brain regions, in categories of sub-cortical (sc) and cortical (c), and CC stands for corpus callosum. A reduction in SNR level was achieved by reducing the number of diffusion encoding directions in each b-shell of the DW-MRI data. The pooled means and standard deviations across participants have been tabulated, along with the coefficient of variation in parentheses. The entries identified in italic under the optimal $b$-value heading were found to be significantly different from the benchmark mean $K^*$ reported in *Table 2*. Sub-optimal result population means were mostly significantly different from the benchmark mean $K^*$, and they are not italicised. The average errors (last column) are relative errors compared to the benchmark kurtosis values reported in *Table 2*.

| | SNR = 6 | SNR = 10 | SNR = 20 | Average error for SNR = 6/10/20 |
|---|---|---|---|---|
| **Optimal _b_-values** | | | | |
| **scGM** | *0.47 ± 0.27(57%)* | 0.65 ± 0.22(33%) | 0.61 ± 0.21(34%) | 37/23/12% |
| Thalamus | *0.57 ± 0.26(46%)* | 0.75 ± 0.20(26%) | 0.71 ± 0.17(24%) | 33/20/11% |
| Caudate | *0.30 ± 0.20(68%)* | 0.46 ± 0.17(38%) | 0.40 ± 0.15(38%) | 43/30/15% |
| Putamen | *0.38 ± 0.21(56%)* | 0.58 ± 0.16(28%) | 0.52 ± 0.14(28%) | 40/28/13% |
| Pallidum | *0.66 ± 0.33(49%)* | 0.90 ± 0.24(26%) | 0.89 ± 0.20(22%) | 39/19/12% |
| **cGM** | 0.40 ± 0.22(54%) | 0.51 ± 0.20(39%) | 0.45 ± 0.18(40%) | 30/36/19% |
| Fusiform | 0.37 ± 0.21(57%) | 0.54 ± 0.20(37%) | 0.45 ± 0.17(38%) | 35/43/20% |
| Lingual | 0.45 ± 0.22(50%) | 0.59 ± 0.19(33%) | 0.52 ± 0.18(34%) | 28/34/18% |
| **WM** | *0.72 ± 0.26(36%)* | 0.91 ± 0.23(26%) | 0.89 ± 0.22(25%) | 23/13/8% |
| Cerebral WM | *0.71 ± 0.25(35%)* | 0.89 ± 0.23(25%) | 0.88 ± 0.22(25%) | 22/13/8% |
| **Sub-optimal _b_-values** | | | | |
| **scGM** | 0.31 ± 0.22(71%) | 0.82 ± 0.29(36) | 0.58 ± 0.25(43%) | 55/45/34% |
| Thalamus | 0.36 ± 0.23(65%) | 0.95 ± 0.25(27%) | 0.72 ± 0.19(26%) | 54/43/31% |
| Caudate | 0.23 ± 0.19(81%) | 0.57 ± 0.17(31%) | 0.40 ± 0.20(52%) | 56/57/40% |
| Putamen | 0.25 ± 0.16(67%) | 0.65 ± 0.17(26%) | 0.44 ± 0.20(46%) | 54/42/36% |
| Pallidum | 0.44 ± 0.29(66%) | 1.27 ± 0.32(25%) | 0.76 ± 0.25(33%) | 56/42/32% |
| **cGM** | 0.25 ± 0.17(67%) | 0.44 ± 0.14(33%) | 0.37 ± 0.17(47%) | 47/30/36% |
| Fusiform | 0.22 ± 0.15(68%) | 0.47 ± 0.16(35%) | 0.33 ± 0.17(52%) | 51/32/38% |
| Lingual | 0.29 ± 0.19(66%) | 0.53 ± 0.16(30%) | 0.39 ± 0.20(51%) | 48/29/40% |
| **WM** | 0.42 ± 0.18(44%) | 1.16 ± 0.36(31%) | 0.72 ± 0.22(31%) | 53/38/28% |
| Cerebral WM | 0.41 ± 0.18(43%) | 1.15 ± 0.35(31%) | 0.74 ± 0.20(28%) | 53/38/27% |
| Cerebellum WM | 0.47 ± 0.21(45%) | 1.27 ± 0.33(26%) | 0.35 ± 0.26(74%) | 57/24/72% |
| CC | 0.69 ± 0.42(61%) | 1.94 ± 0.44(23%) | 0.92 ± 0.19(20%) | 47/102/28% |
| **Cerebellum WM** | *0.83 ± 0.29(35%)* | 1.17 ± 0.25(21%) | 1.15 ± 0.23(20%) | 27/15/11% |
| CC | *0.88 ± 0.36(41%)* | 0.98 ± 0.31(31%) | 0.86 ± 0.25(29%) | 26/17/13% |

Less than $4\,\mathrm{min}$ DW-MRI data acquisitions can potentially replace existing data acquisitions used to obtained DTI metrics, since even the estimation of the apparent diffusion coefficient improves by using DW-MRI data relevant to DKI (*Veraart et al., 2011b*; *Wu and Cheung, 2010*). Additionally, it is increasingly clear that in certain applications the DKI analysis offers a more comprehensive approach for tissue microstructure analysis (*Li et al., 2022b*; *Liu et al., 2022*; *Huang et al., 2022*; *Guo et al., 2022b*; *Goryawala et al., 2023*; *Guo et al., 2022a*; *Wang et al., 2022*; *Li et al., 2022a*; *Maralakunte*

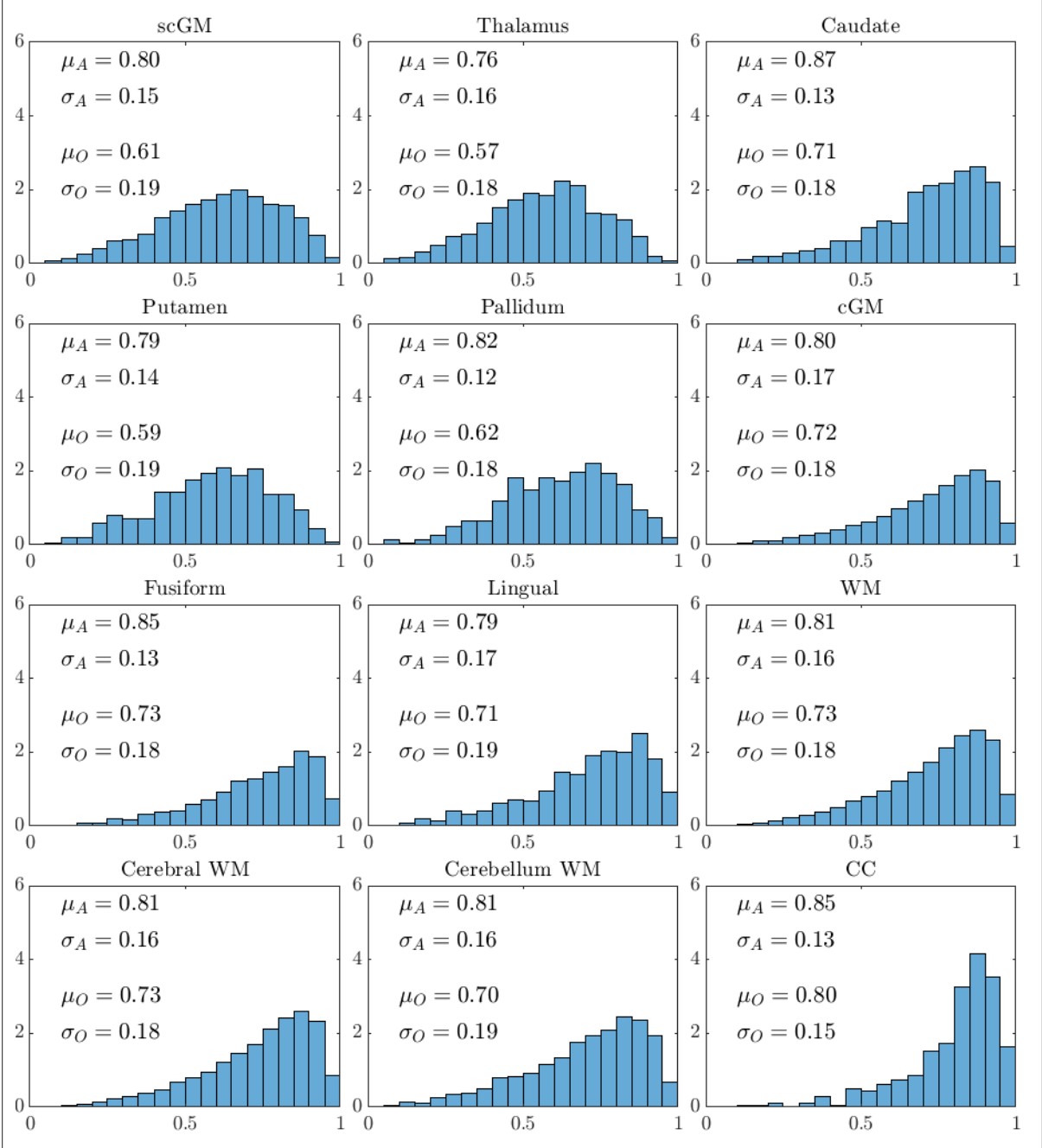

**Figure 10.** Interclass correlation coefficient (ICC) results for mean kurtosis are depicted for the 12 brain regions analysed. The mean ($\mu$) and standard deviation ($\sigma$) computed based on all the Connectome 1.0 DW-MRI data (subscript **A**), and the reduced data achieving an SNR = 10 with optimal four non-zero $b$-value sampling (subscript **O**), are provided for each brain region. Histograms were generated using all data. Mean kurtosis based on the optimised protocol was computed using the sub-diffusion framework using DW-MRI data with the four non-zero $b$-values suggested in *Table 1* and diffusion encoding directions down sampled to achieve an SNR = 10.

*et al., 2023*; *Hu et al., 2022*; *Zhou et al., 2023*; *Li et al., 2022c*). As such, multiple b-shell, multiple diffusion encoding direction DW-MRI acquisitions should be used for the calculation of both DTI and DKI metrics.

### DW-MRI data acquisition considerations

To achieve $R^2 > 0.92$ for estimating kurtosis $K^*$, it is necessary to have four $b$-values, e.g., two relatively small $b$-values (350 s/mm$^2$ using $\Delta = 19$ ms, and 950 s/mm$^2$ using $\Delta = 49$ ms, both with

$G = 68$ mT/m) and two larger $b$-values of around 1500 s/mm$^2$ and 4250 s/mm$^2$ (using $\Delta = 19$ ms and $\Delta = 49$ ms, respectively, both with $G = 142$ mT/m) (see bottom row in **Table 1**). If two or three non-zero $b$-values are considered sufficient (with $R^2 = 0.85$ or 0.90), then the larger $\Delta$ needs to be used to set the largest $b$-value to be 2300 s/mm$^2$, and the other(s) should be set using the smaller $\Delta$. For the two non-zero $b$-values case, the $b$-value from the smaller $\Delta$ should be around 800 s/mm$^2$. For the three non-zero $b$-values case, the $b$-values from the smaller $\Delta$ would then need to be 350 s/mm$^2$ and 1500 s/mm$^2$. Interestingly, $b$-value = 800 s/mm$^2$ lies around the middle of the 350 s/mm$^2$ to 1500 s/mm$^2$ range. The additional gain to $R^2 = 0.92$ can be achieved by splitting the $b$-value with the larger $\Delta$ into two, again with 2300 s/mm$^2$ near the middle of the two new $b$-values set. In addition, the separation between $\Delta_1$ and $\Delta_2$ needs to be as large as plausible, as can be deduced from the simulation result in **Figure 2**, but attention should be paid to signal-to-noise ratio decreases with increased echo times (**He et al., 2022**).

A recent study on optimising quasi-diffusion imaging (QDI) considered $b$-values up to 5000 s/mm$^2$ (**Spilling et al., 2022**). Whilst QDI is a non-Gaussian approach, it is different to the sub-diffusion model, but still uses the Mittag-Leffler function and involves the same number of model parameters. The DW-MRI data used in their study was acquired with a single diffusion time. Nevertheless, particular points are worth noting. Their primary finding was parameter dependence on the maximum $b$-value used to create the DW-MRI data. They also showed the accuracy, precision, and reliability of parameter estimation are improved with increased number of b-shells. They suggested a maximum $b$-value of 3960 s/mm$^2$ for the 4-shell parameter estimation. Our results do not suggest a dependence of the parameter estimates on maximum $b$-value (note, if a maximum $b$-value dependence is present, benchmark versus optimal region specific results in **Tables 2** and **5** should show some systematic difference; and we used the real part of the DW-MRI data and not the magnitude as commonly used). These findings potentially confirm that $\Delta$ separation is an important component of obtaining a quality parameter fit from which mean kurtosis is deduced (**Figure 2**).

## Diffusion gradient pulse amplitudes

Commonly available human clinical MRI scanners are capable of 40 mT/m gradient amplitudes. Recently, the increased need to deduce tissue microstructure metrics from DW-MRI measurements has led to hardware developments resulting in 80 mT/m gradient strength MRI scanners. These were initially sought by research centres. The Connectome 1.0 scanners achieve 300 mT/m gradient amplitudes (**Tian et al., 2022**), which in turn allow large $b$-values within reasonable echo times. The Connectome 2.0 scanner is planned to achieve 500 mT/m gradient amplitudes (**Huang et al., 2021**), providing data for further exploration of the $(q, \Delta)$ space by providing a mechanism for increasing our knowledge of the relationships between the micro-, meso-, and macro-scales. Whilst there are three Connectome 1.0 scanners available, only one Connectome 2.0 scanner is planned for production. Hence, robust and fast methods utilising existing 80 mT/m gradient systems are needed, and the Connectome scanners provide opportunities for testing and validating methods.

The $b$-value is an intricate combination of $\delta$, $G$, and $\Delta$. An increase in any of these three parameters increases the $b$-value (note, $\delta$ and $G$ increase $b$-value quadratically, and $\Delta$ linearly). An increase in $\delta$ can likely require an increase in $\Delta$, the consequence of which is a reduction in signal-to-noise ratio as the echo time has to be adjusted proportionally. Partial Fourier sampling methods aim to counteract the need to increase the echo time by sub-sampling the DW-MRI data used to generate an image for each diffusion encoding direction (**Zong et al., 2021**; **Heidemann et al., 2010**). The ideal scenario, therefore, is to increase $G$, as for the Connectome 1.0 and 2.0 scanners.

Based on our suggested $b$-value settings in **Table 1**, a maximum $b$-value of around 4250 s/mm$^2$ is required for robust mean kurtosis estimation (assume $R^2 > 0.90$ is adequate and achieved via four non-zero $b$-values and two distinct $\Delta$s, and we should highlight that $b$-value = 1500 s/mm$^2$ ($\Delta = 19$ ms) and $b$-value = 4250 s/mm$^2$ ($\Delta = 49$ ms) were achieved using $G = 142$ mT/m). Considering 80 m/Tm gradient sets are the new standard for MRI scanners, an adjustment to $\delta$ to compensate for $G$ is needed (recall, $q = \gamma\delta G$). Hence, for a constant $q$, $G$ can be reduced at the consequence of proportionally increasing $\delta$. This then allows MRI systems with lower gradient amplitudes to generate the relevant $b$-values. For example, changing the $\delta$ from 8 ms to 16 ms would result in halving of $G$ (using a maximum $G$ of 142 m/TM from the Connectome 1.0 can achieve the optimal $b$-values; hence halving would require around 71 m/TM gradient pulse amplitudes). Increasing $\Delta$ above 49 ms to create larger

*b*-value data is unlikely to be a viable solution due to longer echo times leading to a loss in SNR. SNR increases afforded by moving from 3 T to 7 T MRI are most likely counteracted by an almost proportional decrease in $T_2$ times (**Pohmann et al., 2016**), in addition to 7 T MRI bringing new challenges in terms of increased transmit and static field inhomogeneities leading to signal inconsistencies across an image (**Kraff and Quick, 2017**).

## Relationship between sub-diffusion based mean kurtosis $K^*$ and histology

Following **Table 2** (last column), white matter regions showed high kurtosis (0.87 ± 0.22), consistent with a structured heterogeneous environment comprising parallel neuronal fibres, as shown in **Maiter et al., 2021**. Cortical grey matter showed low kurtosis (0.40 ± 0.16). Subcortical grey matter regions showed intermediate kurtosis (0.60 ± 0.21). In particular, caudate and putamen showed similar kurtosis to grey matter, whilst thalamus and pallidum showed similar kurtosis properties to white matter. Histological staining results of the sub-cortical nuclei (**Maiter et al., 2021**) showed the subcortical grey matter was permeated by small white matter bundles, which could account for the similar kurtosis in thalamus and pallidum to white matter. These results confirmed that our proposed sub-diffusion based mean kurtosis $K^*$ is consistent with published histology of normal human brain.

## Time-dependence of diffusivity and kurtosis

The time-dependence of diffusivity and kurtosis has attracted much interest in the field of tissue microstructure imaging. Although our motivation here is not to map time-dependent diffusion, we can nonetheless point out that the assumption of a sub-diffusion model provides an explanation of the observed time-dependence of diffusivity and kurtosis. From (**Equation 4**), the diffusivity that arises from the sub-diffusion model is of the form $D_{SUB} = D_\beta \bar{\Delta}^{\beta-1}$, where $\bar{\Delta}$ is the effective diffusion time, $D_{SUB}$ has the standard units for a diffusion coefficient $\text{mm}^2/\text{s}$, and $D_\beta$ is the anomalous diffusion coefficient (with units $\text{mm}^2/\text{s}^\beta$) associated with sub-diffusion. In the sub-diffusion framework (**Equation 1**), $D_\beta$ and $\beta$ are assumed to be constant, and hence $D_{SUB}$ exhibits a fractional power-law dependence on diffusion time. Then, following (**Equation 8**), $D^*$ is obtained simply by scaling $D_{SUB}$ by a constant $1/\Gamma(1 + \beta)$, and hence also follows a fractional power-law dependence on diffusion time. This time-dependence effect of diffusivity was illustrated in our simulation results, **Figure 4(A) and (C)**.

When it comes to kurtosis, the literature on the time-dependence is mixed. Some work showed kurtosis to be increasing with diffusion time in both white and grey matter (**Aggarwal et al., 2020**), and in grey matter (**Ianus et al., 2021**), whilst others showed kurtosis to be decreasing with diffusion time in grey matter (**Lee et al., 2020**; **Olesen et al., 2022**; **Jelescu et al., 2022**). In this study, we provide an explanation of these mixed results. We construct a simulation of diffusion MRI signal data based on the sub-diffusion model (**Equation 3**) augmented with random Gaussian noise. Then we fit the conventional DKI model to the synthetic data. As shown in **Figure 4(D)**, when there is no noise, $K_{DKI}$ increases with diffusion time in white matter, whilst decreasing with diffusion time in grey matter. When there is added noise, as shown in **Figure 4(B)**, the time-dependency of kurtosis within the timescale of a usual MR experiment is not clear. This goes some way to explaining why the results in the literature on the time-dependence of kurtosis are quite mixed.

Furthermore, we summarise the benefits of using the sub-diffusion based mean kurtosis measurement $K^*$. First, as shown in (**Equation 9**), sub-diffusion based mean kurtosis $K^*$ is not time dependent, and hence has the potential to become a tissue-specific imaging biomarker. Second, the fitting of the sub-diffusion model is straightforward, fast and robust, from which the kurtosis $K^*$ is simply computed as a function of the sub-diffusion model parameter $\beta$. Third, the kurtosis $K^*$ is not subject to any restriction on the maximum *b*-value, as in standard DKI. Hence, its value truly reflects the information contained in the full range of *b*-values.

## Extension to directional kurtosis

The direct link between the sub-diffusion model parameter $\beta$ and mean kurtosis is well established (**Yang et al., 2022**; **Ingo et al., 2014**; **Ingo et al., 2015**). An important aspect to consider is whether mean $\beta$ used to compute the mean kurtosis is alone sufficient for clinical decision making. Whilst benefits of using kurtosis metrics over other DW-MRI data-derived metrics in certain applications are clear, the adequacy of mean kurtosis over axial and radial kurtosis is less apparent. Most studies perform

the mapping of mean kurtosis, probably because the DW-MRI data can be acquired in practically feasible times. Nonetheless, we can point to a few recent examples where the measurement of directional kurtosis has clear advantages. A study on mapping tumour response to radiotherapy treatment found axial kurtosis to provide the best sensitivity to treatment response (*Goryawala et al., 2023*). In a different study, a correlation was found between glomerular filtration rate and axial kurtosis is assessing renal function and interstitial fibrosis (*Li et al., 2022a*). Uniplor depression subjects have been shown to have brain region specific increases in mean and radial kurtosis, whilst for bipolar depression subjects axial kurtosis decreased in specific brain regions and decreases in radial kurtosis were found in other regions (*Maralakunte et al., 2023*). This selection of studies highlights future opportunities for extending the methods to additionally map axial and radial kurtosis.

Notably, estimates for axial and radial kurtosis require directionality of kurtosis to be resolved, resulting in DW-MRI sampling over a large number of diffusion encoding directions within each b-shell (*Jensen and Helpern, 2010*; *Poot et al., 2010*). As such, extension to directional kurtosis requires a larger DW-MRI dataset acquired using an increased number of diffusion encoding directions. The number of b-shells and directions therein necessary for robust and accurate mapping of directional kurtosis based on the sub-diffusion model is an open question.

There are three primary ways of determining mean kurtosis. These include the powder averaging over diffusion encoding directions in each shell, and then fitting the model, as in our approach. A different approach is to ensure each b-shell in the DW-MRI data contains the same diffusion encoding directions, and then kurtosis can be estimated for each diffusion encoding direction, after which the average over directions is used to state mean kurtosis. Lastly, the rank-4 kurtosis tensor is estimated from the DW-MRI data, from which mean kurtosis is computed directly. The latter two approaches are potential candidates for extending to axial and radial kurtosis mapping. Note, in DTI a rank-2 diffusion tensor with six unique tensor entries is needed to be estimated, whilst in DKI in addition to the rank-2 diffusion tensor, the kurtosis tensor is rank-4 with 15 unknowns, resulting in 21 unknowns altogether (*Hansen et al., 2016*). As such, DKI analysis for directional kurtosis requires much greater number of diffusion encoding directions to be sampled than DTI. This automatically means that at least 22 DW-MRI data (including $b$-value = 0) with different diffusion encoding properties have to be acquired (*Jensen and Helpern, 2010*). The traditional approach has been to set five distinct $b$-values with 30 diffusion encoding directions within each b-shell (*Poot et al., 2010*). Hence, to obtain the entries of the rank-4 kurtosis tensor, much more DW-MRI data is needed in comparison to what is proposed for mean kurtosis estimation in this study. Estimation of the tensor entries from this much data is prone to noise, motion and image artifacts in general (*Tabesh et al., 2011*), posing challenges on top of long DW-MRI data acquisition times.

A kurtosis tensor framework based on the sub-diffusion model where separate diffusion encoding directions are used to fit a direction specific $\beta$ is potentially an interesting line of investigation for the future, since it can be used to establish a rank-2 $\beta$ tensor with only six unknowns, requiring at least six distinct diffusion encoding directions. This type of approach can reduce the amount of DW-MRI data to be acquired, and potentially serve as a viable way forward for the combined estimation of mean, axial, and radial kurtosis.

## Kurtosis estimation outside of the brain

Although our study has been focusing on mean kurtosis imaging in the human brain, it is clear that DKI has wide application outside of the brain (*Li et al., 2022b*; *Liu et al., 2022*; *Huang et al., 2022*; *Guo et al., 2022b*; *Li et al., 2022a*; *Zhou et al., 2023*). Without having conducted experiments elsewhere, we cannot provide specific guidelines for mean kurtosis imaging in the breast, kidney, liver, and other human body regions. We can, however, point the reader in a specific direction.

The classical mono-exponential model can be recovered from the sub-diffusion equation by setting $\beta = 1$. For this case, the product between the $b$-value and fitted diffusivity has been reported to be insightful for $b$-value sampling (*Yablonskiy and Sukstanskii, 2010*), in accordance with a theoretical perspective (*Istratov and Vyvenko, 1999*). It was suggested the product should approximately span the (0, 1) range. Considering our case based on the sub-diffusion equation, we can investigate the size of $bD_{SUB}$ by analysing the four non-zero $b$-value optimal sampling regime ($\Delta_1$: 350 s/mm$^2$ and 1500 s/mm$^2$; $\Delta_2$: 950 s/mm$^2$ and 4250 s/mm$^2$ from *Table 1*). Considering scGM, cGM, and WM brain regions alone, the rounded and dimensionless $bD_{SUB}$ values are (0.09, 0.38, 0.20, 0.88), (0.13, 0.55,

0.30, 1.34), and (0.05, 0.23, 0.11, 0.48), respectively, and note that in each case the first two effective sampling values are based on $\Delta_1$, and the other two are derived using $\Delta_2$. Interestingly, the log-linear sampling proposed in *Istratov and Vyvenko, 1999* is closely mimicked by the effective sampling regime (scGM: −2.42, −1.62, −0.97, −0.13; cGM: −2.06, −1.20, −0.60, 0.29; WM: −2.94, −2.23, −1.48, −0.73; by sorting and taking the natural logarithm). This analysis also informs on why it may be difficult to obtain a generally large $R^2$ across the entire brain, since $\beta$ and $D_{SUB}$ are brain region specific and the most optimal sampling strategy should be $\beta$ and $D_{SUB}$ specific. Whilst region specific sampling may provide further gains in the $R^2$ value, and improve ICC values for specific brain regions, such data would take a long time to acquire and require extensive post-processing and in-depth analyses.

## Methods

### Theory

#### Sub-diffusion modelling framework

In biological tissue, the motion of water molecules is hindered by various microstructures, and hence the diffusion can be considerably slower than unhindered, unrestricted, free diffusion of water. The continuous time random walk formalism provides a convenient mathematical framework to model this sub-diffusive behaviour using fractional calculus (*Metzler and Klafter, 2000*). The resulting probability density function $P(x, t)$ of water molecules at location $x$ (in units of mm) at time $t$ (in units of s) satisfies the time fractional diffusion equation:

$$\frac{\partial^\beta P(x,t)}{\partial t^\beta} = D_\beta \nabla^2 P(x,t), \qquad 0 < \beta \leq 1, \tag{1}$$

where $\frac{\partial^\beta}{\partial t^\beta}$ is the time fractional derivative of order $\beta$ ($0 < \beta \leq 1$) in the Caputo sense, $D_\beta$ is the generalised anomalous diffusion coefficient with unit of $mm^2/s^\beta$, and the parameter $\beta$ characterises the distribution of waiting times between two consecutive steps in the continuous time random walk interpretation. When $\beta = 1$, the waiting times have finite mean; when $0 < \beta < 1$, the waiting times have infinite mean, leading to sub-diffusion behaviour. The solution to the time fractional diffusion *Equation 1* in Fourier space is:

$$p(k, t) = E_\beta \left( D_\beta |k|^2 t^\beta \right), \tag{2}$$

where $E_\beta(z) = \sum_{n=0}^{\infty} \frac{z^n}{\Gamma(1 + \beta n)}$ is the single-parameter Mittag-Leffler function, $\Gamma$ is the standard Gamma function and by definition $E_1(z) = \exp(z)$. In the context of diffusion DW-MRI, $k$ in *Equation 2* represents the $q$-space parameter $q = \gamma G \delta$, $t$ represents the effective diffusion time $\bar{\Delta} = \Delta - \delta/3$ and $p(k, t)$ represents the signal intensity $S(q, \bar{\Delta})$, leading to the diffusion signal equation (*Magin et al., 2020*):

$$S(q, \bar{\Delta}) = S_0 E_\beta \left( -D_\beta q^2 \bar{\Delta}^\beta \right). \tag{3}$$

Defining $b = q^2 \bar{\Delta}$, the DW-MRI signal then can be expressed in terms of $b$-values:

$$S(b) = S_0 E_\beta(-bD_{SUB}), \tag{4}$$

where

$$D_{SUB} = D_\beta \bar{\Delta}^{\beta-1} \tag{5}$$

has the standard unit for a diffusion coefficient, s/mm².

#### Diffusional kurtosis imaging

The traditional DKI approach was proposed by Jensen and colleagues (*Jensen et al., 2005*; *Jensen and Helpern, 2010*) to measure the extent of non-Gaussian diffusion in biological tissues using DW-MRI data:

$$S(b) \approx S_0 \exp\left(-bD_{DKI} + \frac{1}{6}b^2 D_{DKI}^2 K_{DKI}\right), \tag{6}$$

where $S$ is the signal for a given diffusion weighting $b$ (i.e., $b$-value), $S_0$ is the signal when $b = 0$, $D_{DKI}$ and $K_{DKI}$ are the apparent diffusivity and kurtosis. A major limitation of (*Equation 6*) is that it was developed based on the Taylor expansion of the logarithm of the signal at $b = 0$, as such $b$-values should be chosen in the neighbourhood of $b$-value = 0 (*Yang et al., 2022*; *Kiselev, 2010*). Hence, to estimate diffusivity and kurtosis, *Jensen and Helpern, 2010* suggested the use of three different $b$-values (such as 0, 1000, 2000 s/mm$^2$) and the maximum $b$-value should be in the range 2000 s/mm$^2$ to 3000 s/mm$^2$ for brain studies. Subsequently, the optimal maximum $b$-value was found to be dependent on the tissue types and specific pathologies, which makes the experimental design optimal for a whole brain challenging (*Chuhutin et al., 2017*). The procedure for fitting kurtosis and diffusivity tensors is also not trivial, and a variety of fitting procedures are currently in use. We refer readers to the descriptive and comparative studies for detail on the implementation and comparison of methods (*Veraart et al., 2011b*; *Chuhutin et al., 2017*).

## Mean kurtosis from the sub-diffusion model

*Yang et al., 2022* established that the traditional DKI model corresponds to the first two terms in the expansion of the sub-diffusion model:

$$S(b) = S_0 E_\beta(-bD_{SUB}) = S_0 \exp\left(-bD^* + \frac{1}{6}b^2 D^{*2} K^* + O(b^3)\right), \tag{7}$$

where diffusivity, $D^*$, and kurtosis, $K^*$, are computed directly via sub-diffusion parameters $D_{SUB}$ and $\beta$:

$$D^* = \frac{D_{SUB}}{\Gamma(1 + \beta)}, \tag{8}$$

$$K^* = 6\frac{\Gamma^2(1 + \beta)}{\Gamma(1 + 2\beta)} - 3, \tag{9}$$

where $D_{SUB}$ is defined in (*Equation 5*). Note the mean kurtosis expression in (*Equation 9*) was also derived by *Ingo et al., 2015* using a different method. Their derivation follows the definition of kurtosis, $K = \langle x^4 \rangle / \langle x^2 \rangle^2 - 3$, i.e., by computing the fourth moment $\langle x^4 \rangle$ and the second moment $\langle x^2 \rangle$ based on the sub-diffusion *Equation 1*.

## Connectome 1.0 human brain DW-MRI data

The DW-MRI dataset provided by *Tian et al., 2022* was used in this study. The publicly available data were collected using the Connectome 1.0 scanner for 26 healthy participants. The first seven subjects had a scan–rescan available. We evaluated qualitatively the seven datasets, and chose the six which had consistent diffusion encoding directions. Subject 2 had 60 instead of 64 diffusion encoding directions, and hence, was omitted from this study. The $2 \times 2 \times 2\,\mathrm{mm}^3$ resolution data were obtained using two diffusion times ($\Delta = 19, 49$ ms) with a pulse duration of $\delta = 8$ ms and $G = 31, 68, 105, 142, 179, 216, 253, 290$ mT/m, respectively, generating $b$-values = 50, 350, 800, 1500, 2400, 3450, 4750, 6000 s/mm$^2$ for $\Delta = 19$ ms, and $b$-values = 200, 950, 2300, 4250, 6750, 9850, 13500, 17800 s/mm$^2$ for $\Delta = 49$ ms, according to $b$-value = $(\gamma\delta G)^2(\Delta - \delta/3)$. 32 diffusion encoding directions were uniformly distributed on a sphere for $b < 2400$ s/mm$^2$ and 64 uniform directions for $b \geq 2400$ s/mm$^2$.

The FreeSurfer's segmentation labels as part of the dataset were used for brain-region-specific analyses. *Tian et al., 2022* provided the white matter averaged group SNR (23.10 ± 2.46), computed from 50 interspersed $b$-value = 0 s/mm$^2$ images for each subject. Both magnitude and the real part of the DW-MRI were provided. Based on an in-depth analysis, the use of the real part of the DW-MRI data was recommended, wherein physiological noise, by nature, follows a Gaussian distribution (*Gudbjartsson and Patz, 1995*).

### Simulated DW-MRI data at specific *b*-values

DW-MRI data were simulated to establish (i) the correspondence between actual versus fitted mean kurtosis using the traditional DKI and sub-diffusion models based on various choices for $\Delta$, and (ii) to investigate the impact of SNR levels and sub-sampling of *b*-values on the mean kurtosis estimate. The DW-MRI signal was simulated using the sub-diffusion model (*Equation 3*) with random Gaussian noise added to every normalised DW-MRI signal instance:

$$S(D_\beta, \beta, q, \bar{\Delta}) = E_\beta(-D_\beta q^2 \bar{\Delta}^\beta) + N(0, \sigma^2), \tag{10}$$

where $N(0, \sigma^2)$ is white noise with mean of zero and standard deviation of $\sigma$ according to the normal distribution.

Two aspects influence $\sigma$ in the case of real-valued DW-MRI data. These include the SNR achieved with a single diffusion encoding direction (i.e., Connectome 1.0 DW-MRI data SNR was derived using only *b*-value = 0 s/mm$^2$ data), and the number of diffusion encoding directions in each *b*-shell across which the powder average is computed:

$$\sigma = \frac{1}{\text{SNR}\sqrt{N_{\text{DIR}}}}, \tag{11}$$

where $N_{\text{DIR}}$ is the number of diffusion encoding directions for each b-shell and assuming it is consistent across b-shells. The $\sigma$ for the Connectome 1.0 data is approximately $1/(23.10 \times 8) = 0.0054$ based on 64 diffusion encoding directions. Achieving of SNR = 5, 10, and 20 for the simulation study can therefore be accomplished by changing only the $\sigma$ and keeping $N_{\text{DIR}} = 64$. As such, $\sigma_{\text{SNR=5}} = 0.0250$, $\sigma_{\text{SNR=10}} = 0.0125$, and $\sigma_{\text{SNR=20}} = 0.0063$.

Three simulation experiments were carried out at various SNR levels. The first simulation experiment was to examine the effect of the number of diffusion times on the accuracy of the parameter fitting for idealised grey matter and white matter cases. In (*Equation 10*) the choices of $D_\beta = 3 \times 10^{-4}$ mm$^2$/s$^\beta$, $\beta = 0.75$, and $D_\beta = 5 \times 10^{-4}$ mm$^2$/s$^\beta$, $\beta = 0.85$, were made for white matter and grey matter, respectively. These two distinct $\beta$s led to $K^*$ of 0.8125 and 0.4733 using (*Equation 9*). Diffusion times were chosen from the range $\Delta_i \in [\delta, \delta + 50]$, where diffusion pulse length was set to $\delta = 8$ ms to match the Connectome 1.0 data. A minimum required separation between any two $\Delta$s was enforced, i.e., $30/(n-1)$, where 30 corresponds with the 30 ms difference between the $\Delta$s for the Connectome 1.0 DW-MRI data, and $n$ is the number of distinct diffusion times simulated. We considered as many as five distinct diffusion times. Simulations were conducted by randomly selecting sets of $\Delta$s for 1000 instances, and then generating individual DW-MRI simulated signals using (*Equation 10*), before fitting for $D_\beta$ and $\beta$, from which $K^*$ was computed using (*Equation 9*). For the case of two diffusion times, suggestions on the separation between them was given based on the goodness-of-fit of the model.

The second simulation experiment was to investigate the effect of the number of diffusion times on the accuracy of parameter fitting. To generate simulated data reflecting observations in human data, $D_\beta$ and $\beta$ were restricted to $D_\beta \in [10^{-4}, 10^{-3}]$ in the unit of mm$^2$/s$^\beta$ and $\beta \in [0.5, 1]$, corresponding to $K^* \in [0, 1.7124]$. For the case of two diffusion times, suggestions on the separation between them were given based on the goodness-of-fit of the model.

The third simulation experiment is to study *b*-value sub-sampling under various SNR levels (SNR = 5, 10, and 20). We set the *b*-values in the simulated data the same as those used to acquire the Connectome 1.0 dataset. At each SNR level, we selected combinations of two, three, and four *b*-values, irrespective of the difference between them and the diffusion time set to generate the *b*-value. Essentially, we explored the entire possible sets of *b*-values for the three regimes, resulting in 120, 560, and 1820 combinations, respectively. A goodness-of-fit measure for model fitting was used to make comparisons between the different *b*-value combinations.

## SNR reduction by downsampling diffusion encoding directions

We performed SNR reduction of the Connectome 1.0 DW-MRI data by downsampling of diffusion directions in each b-shell. The method of multiple subsets from multiple sets (P-D-MM) subsampling algorithm provided in DMRITool (*Cheng et al., 2018*) was applied to the *b*-vectors provided with the Connectome 1.0 DW-MRI data. Note, the b-shells contained 32 directions if $b < 2400$ s/mm$^2$, and 64

directions if $b \geq 2400$ s/mm$^2$. We consider SNR = 20 to be the full dataset. The SNR = 10 data was constructed by downsampling to eight non-collinear diffusion encoding directions in each b-shell, and three were required for the SNR = 6 data. In the downsampled data, each diffusion encoding direction was coupled with the direction of opposite polarity (i.e., SNR = 10 had sixteen measurements for each $b$-value, and SNR = 6 had six).

## Parameter estimation

### Standard DKI model

The maximum $b$-value used to acquire the DW-MRI for standard DKI model fitting is limited to the range 2000 s/mm$^2$ to 3000 s/mm$^2$ due to the quadratic form of (**Equation 6**). We opted to use DW-MRI data generated with $b$-values $= 50, 350, 800, 1500, 2400$ s/mm$^2$ using $\Delta = 19$ ms, and $b$-values $= 200$, $950, 2300$ s/mm$^2$ using $\Delta = 49$ ms. Note, the apparent diffusion coefficient, $D_{DKI}$ in (**Equation 6**) is time dependent, as can be deduced from (**Equation 5**) and (**Equation 8**). Thereby, standard DKI fitting can only be applied to DW-MRI data generated using a single diffusion time. The model in (**Equation 6**) was fitted in a voxelwise manner to the powder averaged (i.e., geometric mean over diffusion encoding directions, often referred to as trace-weighted) DW-MRI data using the `lsqcurvefit` function in MATLAB (Mathworks, Version 2022a) using the trust-region reflective algorithm. Optimisation function specific parameters were set to `TolFun` = 10$^{-4}$ and `TolX` = 10$^{-6}$. Parameters were bounded to the ranges of $D_{DKI} > 0$ and $0 < K_{DKI} \leq 3$.

### Sub-diffusion model

For the single diffusion time case, the sub-diffusion model in (**Equation 3**) was fitted to the powder averaged DW-MRI data in a voxelwise manner using the same MATLAB functions as in the previous section. For each subject, spatially resolved maps of $D_\beta$ and $\beta$ were generated. The fitting strategy for multiple diffusion time data is to solve:

$$(D_\beta, \beta) = argmin \sum_{\substack{i=1,2,\ldots,n, \\ j=1,2,\ldots,J_i}} \left[ S_{ij} - SUB\left(\bar{\Delta}_i, q_j; D_\beta, \beta\right) \right]^2 \tag{12}$$

where $S_{ij}$ is the signal at the $i$th effective diffusion time $\bar{\Delta}_i$ and the $j$th q-space parameter $q_j$, $SUB$ is the sub-diffusion model (**Equation 3**) at $\bar{\Delta}_i$ and $q_j$ for a given set of $(D_\beta, \beta)$, $n$ is the number of diffusion times, and $J_i$ is the number of $q$-values corresponding to $\bar{\Delta}_i$ in data acquisition. This objective function allows incorporation of an arbitrary number of diffusion times, each having arbitrary number of $q$-values. Parameters were bounded to the ranges of $0 < \beta \leq 1$ and $D_\beta > 0$ for all the parameter fittings. Model parameters were found to be insensitive to the choice of initial values. Parameters $D^*$ and $K^*$ were computed analytically using the estimated $D_\beta$ and $\beta$ according to (**Equation 8**) and (**Equation 9**).

## Goodness-of-fit and region-based statistical analysis

For all of the simulation results, the coefficient of determination, referred to as $R^2$, was used to assess how well the mean kurtosis values were able to be fitted. This value was calculated using a standard $R^2$ formula

$$R^2 = 1 - \frac{\sum_i (K_{\text{Simulated},i} - K_{\text{Fitted},i})^2}{\sum_i (K_{\text{Simulated},i} - \bar{K}_{\text{Simulated}})^2} \tag{13}$$

where $\bar{K}_{\text{Simulated}}$ is the mean simulated $K$ value.

For human data, we computed the region specific mean and standard deviation for each subject, and reported the weighted mean and pooled standard deviation along with the coefficient of variation (CV), defined as the ratio of the standard deviation to the mean. The weights were the number of voxels in the associated regions in each subject. Following *Barrick et al., 2020*, the tissue contrast (TC) is computed as $TC = \left|\mu_{WM} - \mu_{GM}\right| / \sqrt{\sigma_{WM}^2 + \sigma_{GM}^2}$, where $\mu_{WM}$ and $\mu_{GM}$ are the mean parameter values in white and grey matters; and $\sigma_{WM}$ and $\sigma_{GM}$ are the standard deviations of parameter values. Higher TC values indicate greater tissue contrast.

Human data were analysed voxelwise, and also based on regions-of-interest. We considered three categories of brain regions, namely sub-cortical grey matter (scGM), cortical grey matter (cGM), and white matter (WM). The scGM region constituted the thalamus (FreeSurfer labels 10 and 49 for left and right hemisphere), caudate (11, 50), putamen (12, 51), and pallidum (13, 52). The cGM region was all regions (1000–2999) and separately analysed the fusiform (1007, 2007) and lingual (1013, 2013) brain regions, whilst the WM had white matter fibre regions from the cerebral (2, 41), cerebellum (7, 46), and corpus callosum (CC; 251–255) areas. The average number of voxels in each region were 3986 (scGM), 53326 (cGM), 52121 (WM), 1634 (thalamus), 831 (caudate), 1079 (putamen), 443 (pallidum), 2089 (fusiform), 1422 (lingual), 48770 (cerebral WM), 2904 (cerebellum WM), and 447 (CC). For each brain region, a *t*-test was performed to test for significant differences in mean kurtosis population means under the optimal and sub-optimal *b*-value sampling regimes presented in *Table 5* compared to the benchmark parameter values presented in *Table 2*.

### Scan–rescan analysis using intraclass correlation coefficient (ICC)

For each of the six subjects, both the first (scan) and second (rescan) scan images were registered to the first scan images of Subject 1 using inbuilt MATLAB (Mathworks, Version 2022a) functions (`imregtform` and `imwarp`). We used 3D affine registration to account for distortions and warps common in DW-MRI data. Cubic spline interpolation was applied to resample both the scan and rescan DW-MRI data for each subject onto Subject 1's first scan data grid. The FreeSurfer labels for Subject 1's first scan were used for brain region analysis. For each voxel, the ICC measure was applied to assess scan–rescan reproducibility of mean kurtosis, as described by *Duval et al., 2018* and *Fan et al., 2021*,

$$\text{ICC} = \frac{s_{\text{inter}}^2}{s_{\text{intra}}^2 + s_{\text{inter}}^2} \tag{14}$$

with $s_{\text{intra}}^2 = \text{mean}_i\left(1/2 \cdot ((m_{i,scan} - \bar{m}_i)^2 + (m_{i,rescan} - \bar{m}_i)^2)\right)$ and $s_{\text{inter}}^2 = \text{var}_i(\bar{m}_i)$. Here, $m_{i,scan}$ is the kurtosis value measured in the voxel for subject $i$, and $\bar{m}_i$ is the average value between scan and rescan. An ICC histogram and the mean and standard deviation descriptive statistics were generated for all brain regions analysed.

## Conclusion

The utility of DKI for inferring information on tissue microstructure was described decades ago. Continued investigations in the DW-MRI field have led to studies clearly describing the importance of mean kurtosis mapping to clinical diagnosis, treatment planning and monitoring across a vast range of diseases and disorders. Our research on robust, fast, and accurate mapping of mean kurtosis using the sub-diffusion mathematical framework promises new opportunities for this field by providing a clinically useful, and routinely applicable mechanism for mapping mean kurtosis in the brain. Future studies may derive value from our suggestions and apply methods outside the brain for broader clinical utilisation.

## Acknowledgements

Qianqian Yang and Viktor Vegh acknowledge the financial support from the Australian Research Council (ARC) Discovery Project scheme (DP190101889) for funding a project on mathematical model development and MRI-based investigations into tissue microstructure in the human brain. Qianqian Yang also acknowledges the support from the ARC Discovery Early Career Research Award (DE150101842) for funding a project on new mathematical models for capturing heterogeneity in human brain tissue. Authors also thank the members of the Anomalous Relaxation and Diffusion Study (ARDS) group for many interesting discussions involving diffusion MRI.

## Additional information

### Funding

| Funder | Grant reference number | Author |
|---|---|---|
| Australian Research Council | Discovery Early Career Research Award DE150101842 | Qianqian Yang |
| Australian Research Council | Discovery Project Award DP190101889 | Qianqian Yang Viktor Vegh |

The funders had no role in study design, data collection, and interpretation, or the decision to submit the work for publication.

### Author contributions

Megan E Farquhar, Software, Formal analysis, Validation, Investigation, Visualization, Methodology, Writing – original draft, Writing – review and editing; Qianqian Yang, Viktor Vegh, Conceptualization, Software, Formal analysis, Supervision, Funding acquisition, Validation, Investigation, Methodology, Writing – original draft, Project administration, Writing – review and editing

### Author ORCIDs

Megan E Farquhar ⓘ https://orcid.org/0000-0003-0647-3637
Qianqian Yang ⓘ https://orcid.org/0000-0002-4331-5059

### Ethics

This study uses a publicly available MGH Connectome Diffusion Microstructure Dataset. The institutional review board of Mass General Brigham approved the study protocol and ensured that all relevant ethical regulations were complied with. Written informed consent was obtained from all participants. More details about the MGH Connectome Diffusion Microstructure Dataset are available at https://doi.org/10.1038/s41597-021-01092-6.

Reviewer #1 (Public review): https://doi.org/10.7554/eLife.90465.3.sa1
Reviewer #2 (Public review): https://doi.org/10.7554/eLife.90465.3.sa2
Author response https://doi.org/10.7554/eLife.90465.3.sa3

## Additional files

### Supplementary files

• MDAR checklist

### Data availability

The Connectome 1.0 human brain DW-MRI data used in this study is part of the MGH Connectome Diffusion Microstructure Dataset (CDMD) (*Tian et al., 2022*), which is publicly available on the figshare repository: https://doi.org/10.6084/m9.figshare.c.5315474. MATLAB codes generated for simulation study, parameter fitting, and optimisation of *b*-value sampling are openly available at https://github.com/m-farquhar/SubdiffusionDKI (copy archived at *Farquhar, 2023*).

The following previously published dataset was used:

| Author(s) | Year | Dataset title | Dataset URL | Database and Identifier |
|---|---|---|---|---|
| Tian Q, Fan Q, Witzel T, Polackal MN, Ohringer NA, Ngamsombat C, Russo AW, Machado N, Brewer K, Wang F, Setsompop K, Polimeni JR, Keil B, Wald LL, Rosen BR, Klawiter E, Nummenmaa A, Huang SY | 2022 | Comprehensive diffusion MRI dataset for in vivo human brain microstructure mapping using 300 mT/m gradients | https://doi.org/10.6084/m9.figshare.c.5315474 | figshare, 10.6084/m9.figshare.c.5315474 |

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
