## [Editor Report · eLife assessment]

This paper proposes a **valuable** new method for the assessment of the mean kurtosis for diffusional kurtosis imaging by utilizing a recently introduced sub-diffusion model. The evidence supporting the claims that this technique is robust and accurate in brain imaging is **solid**; however, there is a need to include a summary of the clear limitations.

---

## [Referee Report · Reviewer #1 (Public review)]

This study introduces an innovative method for assessing the mean kurtosis, utilizing the mathematical foundation of the sub-diffusion framework. In particular, a new fitting technique that incorporates two different diffusion times is proposed to estimate the parameters of the sub-diffusion model. The evaluation of this technique, which generates kurtosis maps based on the sub-diffusion framework, is conducted through simulations and the examination of data obtained from human subjects.

The authors have revised the manuscript to address the initial critiques. However, there appears to be some confusion regarding the following responses.

"The comment "... using the new sub-diffusion model -an approximation of the DKI-based signal expression..." is a bit misleading. In fact we propose that the reverse interpretation is the more suitable way to view the relationship: the DKI model is a degree-2 approximation of the sub-diffusion model, as in eq. (7).

We appreciate the suggestion. However, unfortunately, it is not appropriate to generate data with the DKI model, as the maximum b-value is limited to 2000~3000s/mm^2 and hence the DKI model cannot represent diffusion MRI signals from a full spectrum of b-values. A key strength of our proposed model is that it removes this limitation. "

The main motivation of this study is to investigate the feasibility of the sub-diffusion model, which was proposed in Yang et al., NeuroImage 2022, to provide fast and robust estimation of kurtosis model parameters. I understand that mathematically, the DKI model can be written as a degree two approximation of the sub-diffusion model. However, the hypothesis is that the proposed sub-diffusion model can be used to obtain practically useful mapping of mean kurtosis. Therefore, unless the authors use a different parameter or phenomenon as the "true" or "ground-truth kurtosis," this study examines whether the sub-diffusion model parameters can serve as an approximation to the conventional DKI parameters.

With the current simulation study design, (1) the data is generated by the proposed sub-diffusion model, (2) the "ground-truth" or "true" D* and K* are computed based on the proposed equality (Eq.7); (3) and then the data is fit with the conventional DKI model and also with the proposed sub-diffusion model. Since the data is generated by the proposed model, and the ground truth (or true) values calculated by the proposed equality, as expected, the fitted kurtosis values by the sub-diffusion model match better with the simulated ones compared to the conventional DKI model.

Furthermore, as the authors noted, the sub-diffusion model eliminates the restriction on b-value selection, allowing for DWI data acquisition with higher b-values. However, it is unclear how the new K* and D* values, calculated directly from the sub-diffusion model using a higher b-value DWI protocol, are superior to the K and D values from the conventional DKI model, which uses a DWI protocol limited to b-values of 2000-3000 s/mm². In clinical practice, b-values of 2000-3000 s/mm² are generally considered "high b-value."

---

## [Referee Report · Reviewer #2 (Public review)]

Summary:

The authors present an interesting technique for analysis of diffusion magnetic resonance images (dMRI) using a sub-diffusion model of the diffusion process. They show that the results of their technique when fitted to dMRI with two diffusion times provide robust diffusion coefficient and kurtosis measures.

Strengths:

The measures provided by the sub-diffusion technique are robust and can be reliably estimated from short dMRI data acquisitions. This is potentially useful in application to clinical studies.

Weaknesses:

The authors do not fully demonstrate that their D* and K* measures are not affected by diffusion time. Potential limitations of the technique are not considered.

This reviewer suggests that the paper would benefit from considerations of the limitations of the applied techniques. This would include consideration of:

(i) The use of the sub-diffusion model in the simulation studies - there are circular arguments that should be considered.

(ii) The time dependence of D* and K*. This is because the human data provided in Tables 3 and 4 (for Δ=19ms and Δ=49ms) seem to show that

D* and K* are time dependent.

With respect to the second point this reviewer acknowledges the authors' argument that when the fitting is performed over the higher dimensional space that includes multiple diffusion times then this leads to a more robust estimation of sub-diffusion measures. However, the authors only include two diffusion times in their in-vivo human analysis (Δ=19ms and Δ=49ms) so it is not possible for them to show here that different pairs of diffusion times lead to invariant D* and K* values. This is a limitation of the study as the authors show there is time dependence of D* and K* in tables 3 and 4 (when the model is fitted to single diffusion times). Potentially the larger apparent time dependence of K* in white matter compared to grey matter (tables 3 and 4) could lead to the tissue specific differences in root mean squared error shown in Figure 7.

This reviewer requests that the authors discuss their results more clearly with respect to these potential limitations and include some discussion of their single (and multiple) diffusion time results (for D_SUB and K*) in comparison with the time dependent DKI literature.

---

## [Author Response]

The following is the authors’ response to the original reviews.

**eLife assessment**
This paper proposes a valuable new method for the assessment of the mean kurtosis for diffusional kurtosis imaging by utilizing a recently introduced sub-diffusion model. The evidence supporting the claims that this technique is robust and accurate in brain imaging is incomplete. The work could be of interest in the research and clinical arena.

We thank the editors for their assessment and the reviewers for their careful reading and feedback that helped to improve the manuscript. We have addressed all the reviewers’ concerns and would like to request an update of the assessment to reflect the revisions we have made.

Below, we address the reviewers’ comments.

**Public Reviews:**

**Reviewer #1 (Public Review):**
Summary:This study introduces an innovative method for assessing the mean kurtosis, utilizing the mathematical foundation of the sub-diffusion framework. In particular, a new fitting technique that incorporates two different diffusion times is proposed to estimate the parameters of the sub-diffusion model. The evaluation of this technique, which generates kurtosis maps based on the sub-diffusion framework, is conducted through simulations and the examination of data obtained from human subjects.

We thank Reviewer #1 for pointing out the novelty and innovation of our work.

Strengths:The utilization of the sub-diffusion model for tissue characterization is a significant conceptual advancement for the field of diffusion MRI. This study adeptly harnesses this approach for an accurate estimation of the parameters of the widely employed diffusion model, DKI, leveraging their established analytical interconnection as evidenced in prior research. Notably, this approach not only proposes a robust, fast, and accurate technique for DKI parameter estimation but also underscores the viability of deploying the sub-diffusion model for tissue characterization, substantiated by both simulated and human subject analyses. The paper is very-well written; well-organized; and coherent. The simulation study included different aspects of water diffusion as captured by diffusion-weighted MRI such as varying diffusion times and different b-value subpopulations, resulting in a comprehensive and thorough discussion.

We thank Reviewer #1 for highlighting the the strengths of our work.

Weaknesses:The primary objective of this study is to demonstrate a robust approach for estimating DKI parameters by directly calculating them using the parameters of the sub-diffusion model. This premise, however, relies on the assumption that the sub-diffusion model effectively characterizes the diffusion MRI signal and that its parameters are both robust and accurate. Throughout the manuscript, the term "ground truth kurtosis K" is frequently used to denote the "true K" value in the context of the simulation study. Nonetheless, given that the data is simulated using the new sub-diffusion model - an approximation of the DKI-based signal expression- this value cannot truly be considered the "ground truth K". The simulation study highlights the robustness and accuracy of D* and K*, but it inherently operates under the assumption that the observed data is in the form of the sub-diffusion model.

It is correct that our study operates under the assumption that the observed data is in the form of the sub-diffusion model, and indeed one of the key outcomes of this work is to demonstrate the effectiveness of that assumption and the new possibilities it brings. Naturally, using any mathematical model at all carries assumptions. Over the past two decades, many mathematical and biophysical models have been proposed to characterise diffusion MRI signals. However, model validation remains an open challenge in the field. In this, as well as in our previous work (Yang et al, NeuroImage, 2022), we have shown that our proposed sub-diffusion model not only provides a much better fitting compared to the traditional DKI method, overcoming the major limitation of the traditional DKI method on the maximum b-value, but also generates brain maps with superior tissue contrast and elucidates previously unseen structure.

We have replaced the term “ground truth kurtosis K” with “true kurtosis K”.

The comment “… using the new sub-diffusion model – an approximation of the DKI-based signal expression…” is a bit misleading. In fact we propose that the reverse interpretation is the more suitable way to view the relationship: the DKI model is a degree-2 approximation of the sub-diffusion model, as in eq. (7).

**Reviewer #2 (Public Review):**
Summary: The authors present a technique for fitting diffusion magnetic resonance images (dMRI) to a sub-diffusion model of the diffusion process within brain imaging. The authors suggest that their technique provides robust and accurate calculation of diffusional kurtosis imaging parameters from which high quality images can be calculated from short dMRI data acquisitions at two diffusion times.Strengths: If the authors can show that the dMRI signal in brain tissue follows a sub-diffusion model decay curve then their technique for accurately and robustly calculating diffusional kurtosis parameters from multiple diffusion times would be of benefit for tissue microstructural imaging in research and clinical arenas.

In Figure 7, we showed that the diffusion MRI signals follow the sub-diffusion model decay curves.

Weaknesses: The applied sub-diffusion model has two parameters that are invariant to diffusion time, D_β and β which are used to calculate the diffusional kurtosis measures of a diffusion time dependent D* and a diffusion time invariant K*. However, the authors do not demonstrate that the D_β, β and K* parameters are invariant to diffusion time in brain tissue.

In our proposed sub-diffusion model, D_β and β are assumed to be time-independent parameters, which is a key strength of the approach. The goal is to characterise tissue-specific properties (D_β for diffusivity and β for the extent of tissue complexity) that do not rely on the diffusion time setting in diffusion MRI experiments. To extract such time-independent properties, we proposed a new sampling and fitting strategy – fitting at least two diffusion time data together.

The authors' results visually show that there is time dependence of the K* measure (in Figure 6) that is more apparent in white matter with K* values being higher for diffusion times of ∆=49 ms than ∆ = 19 ms. The diffusion time dependence of K* indicates there is also diffusion time dependence of β.

The discrepancies in the fitted K* for ∆ = 19 ms and ∆ = 49 ms separately do not necessarily imply that there is a true time dependence in these parameters. Rather, this can be explained by a deficiency of data when fitting a two-dimensional surface (S is a function of q and ∆) based on data along a single curve for a fixed value of ∆. Without properly sampling the surface across two independent coordinates, one cannot expect a fully reliable fit. Indeed, a great advantage of our proposed method is to allow fitting data with multiple values of ∆, and thereby getting a richer data set with which to fit the full signal surface S(q, ∆). The results for fitting ∆ = 19 ms and ∆ = 49 ms data together clearly show the benefits of this approach, with superior contrast achieved.

Furthermore, Figure 7 shows that there is a tissue specific root mean squared error in model fitting over the two diffusion times which indicates greater deviation from the model fit in white matter than grey matter.

Although the errors are not completely tissue-independent, please note the magnitude of the RMSE is very small. The quality of the fitting in both white and grey matter is shown in sub-figures (A)-(H) for several representative voxels.

To show that the sub-diffusion model is robust and accurate (and consequently that K* is robust and accurate) the authors would have to demonstrate that there is no diffusion time-dependence in both D_β and β in application to brain imaging data for each diffusion time separately. Simulated data should not be used to demonstrate the robustness and accuracy of the sub-diffusion model or to determine optimization of dMRI acquisition parameters without first demonstrating that D_β and β are invariant to diffusion time. This is because simulated signals calculated by using the sub-diffusion characteristic equation of dMRI signal decay will necessarily have diffusion time invariant D_β and β parameters. Without further information demonstrating diffusion time invariance of D_β, β and K* it is not possible to determine whether the authors have achieved their aims or that their results support their conclusions.

First, as explained above, the dMRI signal S is a function of q and ∆, i.e., a two-dimensional surface S(q, ∆), and hence fitting data sampled from single diffusion time (i.e., one curve on the surface) cannot provide reliable parameters, as seen in the discrepancies in K* in Figure 6 (bottom two rows). Our proposed new sampling and fitting strategy overcomes this issue. That is, to obtain a reliable fitting, one should fit data from at least two diffusion times together (i.e., sampling data from at least two curves on the signal surface).

Second, to demonstrate that D_β and β are time invariant, one would require data at several diffusion times with high b values. Such data cannot be easily obtained. The data used in this current study is the MGH Connectome 1.0 human brain data, which only contains two diffusion times, ∆ = 19 ms and ∆ = 49 ms.

Hence, we conducted numerical experiments to demonstrate our idea. In Figure 3, we showed that (i) the variability of the fitted parameters is significantly reduced when moving from fitting single diffusion time data to two diffusion time data, and (ii) the difference in fitting three diffusion times compared to two is very minor, indicating convergence towards the correct time-independent parameter values. The results from fitting human brain data (Figure 6 and Tables 2-4) agree with the expectations from our numerical experiments. Hence, we believe that we have provided sufficient evidence to support our proposed sub-diffusion model and its optimal fitting strategy.

**Recommendations for the authors:**

**Reviewer #1 (Recommendations For The Authors):**
It is clear that the authors preferred generating the data by using sub-diffusion model's signal expression as it has many benefits, such as allowing different diffusion times to be incorporated, and hence investigation of the effect of the number of diffusion times on the accuracy of the parameter fitting. I recommend adding another simulation study by generating the data with the DKI model expression (as the goal of the study is to provide an accurate mapping of diffusional mean kurtosis), fitting the data to the sub-diffusion model's expression in Eq. (10), and then calculating K* and D* by Eqs. (8) and (9) only for a fixed diffusion time and one b-value subset.

We appreciate the suggestion. However, unfortunately it is not appropriate to generate data with the DKI model, as the maximum b-value is limited to 2000~3000s/mm^2 and hence the DKI model cannot represent diffusion MRI signals from a full spectrum of b-values. A key strength of our proposed model is that it removes this limitation.

There is a typo on Page 24, Line 581; "b<=2400" should be b>=2400.

We have fixed this typo.

**Reviewer #2 (Recommendations For The Authors):**
As the authors state the sub-diffusion model has two parameters, D_β and β that are invariant to diffusion time, and give rise to a time-varying diffusion coefficient in mm^2s^-1 and a time invariant kurtosis. However, there is a need to be clearer and more specific about the implications of the sub-diffusion model. The manuscript would be improved by the authors:(a) Defining the time-varying diffusion coefficient that arises from the model, its functional form and properties.

We refer Reviewer#2 to eq.(5) and eq.(8) for the definition of time-varying diffusion coefficients D* and D_SUB and their relationship.

(b) Clearly discuss the implications of this with respect to other time-varying diffusion coefficient methods in the current literature.

We refer Reviewer#2 to the section “Time-dependence of diffusivity and kurtosis” under “Discussions”.

(c) Demonstrating that D_β and β do not vary with diffusion time when estimated from dMRI acquired on human participants.

We have addressed this comment in the public review.

The manuscript would benefit from increases in clarity in all sections and the authors identifying typographical errors.

We have updated the relevant text in the revised manuscript to make it clearer, including fixing typos.

Specific improvements to clarity in the methods and results section would include:Line 620: Why were parameter approximations for model fitting to simulated data restricted to the ranges D_β∈[10^(-4),10^(-3)] and β∈[0.5,1] but in fitting to brain imaging data the ranges were D_β>0 and 0<β<=1.

The parameter ranges for model fitting to both the simulated and human data were set to the same: D_β>0 and 0<β<=1. To generate simulated data, D_β and β ranges were restricted to reflect observations in human brain data. We have updated the text to make this clearer.

Lines 622, 628 & 629: Which goodness of fit measure was used?

The goodness of fit measure for all simulated results is the coefficient of determination, or R^2 value, as noted in the “Goodness-of-fit and region-based statistical analysis” section under Methods. We have updated the text to make this clearer.

Line 666: The method for computation of R^2 within the coefficient of determination should be stated as there are several ways of calculating an R^2 value.

The formula for computing R^2 has been added to the text.

Line 685: A t-test is mentioned but it is not clear as to the inputs to this test, or where the results of this analysis are presented.

We have updated the text to make this clearer. The results of this analysis are presented in Table 5. The entries identified in italic under the optimal b-value heading were found to be significantly different from the benchmark mean K* reported in Table 2.

Line 696: It is not clear how the intra-class correlation coefficient histograms are computed from six subjects. This applies to results in Figure 10 that require greater clarity in the description.

The formula for computing the intra-class correlation coefficient has been added to the sub-section “Scan-rescan analysis using intraclass correlation coefficient (ICC)” under “Methods”.

It would be helpful if the authors primarily report results pertaining to the model parameters D_β and β. This is because D* and K* are calculated from D_β and β. Conditions for robust and accurate estimation of D_β and β will provide robust and accurate measures for D* and K*.

Two new tables for the model parameters D_β and β have been added. Please see Tables 3 and 4 in the revised manuscript.

The authors state that fitted model parameters are not affected by maximum b-value (paragraph beginning line 366). This statement is based on their model simulation results. Could the authors provide data to support this based on the application of their model to the human brain imaging data?

We would like to clarify that our statement is indeed based on human brain imaging. As stated in the paragraph beginning line 366, both results in Table 2 (using full dataset) and Table 5 (using dataset with optimal b-value sampling) are generated from the Connectome human brain data. If maximum b-value dependence is present, benchmark (Table 2) versus optimal region-specific results (Table 5, or previously Table 3) should show some systematic difference.